# Adaptive Neighborhood-Constrained Q Learning for Offline Reinforcement Learning

**Yixiu Mao**[1], **Yun Qu**[1], **Qi Wang**[1], **Xiangyang Ji**[1]
[1]Department of Automation, Tsinghua University
myx21@mails.tsinghua.edu.cn, xyji@tsinghua.edu.cn

## Abstract

Offline reinforcement learning (RL) suffers from extrapolation errors induced by out-of-distribution (OOD) actions. To address this, offline RL algorithms typically impose constraints on action selection, which can be systematically categorized into density, support, and sample constraints. However, we show that each category has inherent limitations: density and sample constraints tend to be overly conservative in many scenarios, while the support constraint, though least restrictive, faces challenges in accurately modeling the behavior policy. To overcome these limitations, we propose a new neighborhood constraint that restricts action selection in the Bellman target to the union of neighborhoods of dataset actions. Theoretically, the constraint not only bounds extrapolation errors and distribution shift under certain conditions, but also approximates the support constraint without requiring behavior policy modeling. Moreover, it retains substantial flexibility and enables pointwise conservatism by adapting the neighborhood radius for each data point. In practice, we employ data quality as the adaptation criterion and design an adaptive neighborhood constraint. Building on an efficient bilevel optimization framework, we develop a simple yet effective algorithm, Adaptive Neighborhood-constrained Q learning (ANQ), to perform Q learning with target actions satisfying this constraint. Empirically, ANQ achieves state-of-the-art performance on standard offline RL benchmarks and exhibits strong robustness in scenarios with noisy or limited data.

## 1 Introduction

Reinforcement learning (RL) tackles sequential decision-making problems and has gained considerable attention in recent years [58, 70, 67, 14]. Despite its promise, RL faces practical challenges, notably the high data collection costs [38] and exploration risks [24]. Offline RL offers a compelling alternative by learning from a static dataset collected by a behavior policy [43, 44]. It enables the use of existing large-scale datasets [33, 52, 63] and reduces the dangers of unsafe exploration. However, it also introduces a key challenge: evaluating out-of-distribution (OOD) actions leads to extrapolation errors [23], which further causes value overestimation and significant performance degradation [44].

To address this issue, offline RL approaches typically impose constraints on the action selection process. A common strategy is to align the probability densities of the trained and behavior policies [83, 21], enforcing a *density constraint*. This is usually achieved using divergence metrics such as reverse Kullback-Leibler (KL) [83, 32], forward KL (i.e., behavior cloning) [21, 61], Fisher [39], or the implicit CQL divergence [42]. While straightforward, these methods can be overly restrictive both theoretically and empirically [41], in cases where the overall quality of the behavior policy is low. To overcome this limitation, recent work has explored the most relaxed *support constraint*, which only keeps the selected actions within the support of the behavior policy [82, 53, 93]. Accomplishing this generally necessitates high-fidelity estimation of the behavior policy through advanced generative modeling techniques [26, 82, 93, 9]. However, such modeling is challenging due to the

39th Conference on Neural Information Processing Systems (NeurIPS 2025).

Table 1: A brief summary of constraint types in offline RL research.

| Constraint type | Description | Algorithms | Key characteristics |
|---|---|---|---|
| Density | Enforce density proximity between the trained and behavior policies | BRAC [83], TD3BC [21], CQL [42] | Straightforward but heavily limited by the overall quality of behavior policy |
| Sample | Restrict action selection to dataset actions | IQL[40], XQL[25], SQL[88] | Avoid extrapolation error but lack action generalization beyond the dataset |
| Support | Restrict action selection to behavior policy's support | BCQ[23], BEAR[41], SPOT [82] | Least restrictive but require accurate behavior policy modeling |
| Neighborhood | Restrict action selection to certain neighborhoods of dataset actions | ANQ (Ours) | Flexible and approximate support constraint without behavior modeling |

high-dimensional and multi-modal nature of real-world data [93], subjecting these methods to heightened error susceptibility and increased computational overhead. Alternatively, the *sample constraint* has emerged, formulating the Bellman target exclusively using actions in the dataset [6, 40, 92, 88]. These methods are easy to implement and effectively avoid extrapolation errors [40]. However, their performance is inherently limited by a lack of action generalization beyond the offline dataset, often resulting in overly conservative policies when near-optimal actions are rare in the dataset.

This work aims to address the over-conservatism of the density and sample constraints while avoiding complex behavior modeling required by the support constraint. To this end, we introduce a new *neighborhood constraint* that restricts action selection in the Bellman target to the union of neighborhoods of dataset actions. Theoretically, the constraint not only bounds extrapolation errors and distribution shift under certain conditions, but also approximates the least restrictive support constraint without behavior modeling. Moreover, it maintains high flexibility and can achieve pointwise conservatism by adapting the neighborhood radius for each data point. In light of real-world data patterns, we adopt data quality as the adaptation criterion and develop an adaptive neighborhood constraint in practice. It assigns larger neighborhood radii to low-advantage dataset actions to promote a broader search, and smaller neighborhood radii to high-advantage dataset actions to limit the overall extrapolation error.

To enforce the proposed constraint, we introduce an efficient bilevel optimization framework and, based on it, develop a simple yet effective algorithm, Adaptive Neighborhood-constrained Q learning (ANQ), which performs Q learning with target actions constrained accordingly. Specifically, in the inner optimization, we maximize the Q function separately within each dataset action's neighborhood; in the outer optimization, we implicitly maximize the Q function over all available neighborhoods via expectile regression [40]. With the Q function being trained, the policy is independently extracted by weighted regression toward optimized actions within the neighborhoods, obtained in the inner maximization. Empirically, ANQ achieves state-of-the-art performance on standard offline RL benchmarks [20], including Gym locomotion tasks and challenging AntMaze tasks. Moreover, benefiting from the flexible constraint without behavior modeling errors, ANQ attains superior performance in both noisy and limited data scenarios compared to algorithms with other types of constraints. The code is available at https://github.com/thu-rllab/ANQ.

## 2 Preliminaries

**RL.** In RL, the environment is typically modeled as a Markov Decision Process (MDP) $\mathcal{M} = (\mathcal{S}, \mathcal{A}, P, R, \gamma, d_0)$, with state space $\mathcal{S}$, action space $\mathcal{A}$, transition dynamics $P : \mathcal{S} \times \mathcal{A} \to \Delta(\mathcal{S})$, reward function $R : \mathcal{S} \times \mathcal{A} \to [0, R_{\max}]$, discount factor $\gamma \in [0, 1)$, and initial state distribution $d_0$ [73]. The agent seeks a policy $\pi : \mathcal{S} \to \Delta(\mathcal{A})$ that maximizes the expected return:

$$\eta(\pi) = \mathbb{E}_{s_0 \sim d_0, a_t \sim \pi(\cdot|s_t), s_{t+1} \sim P(\cdot|s_t, a_t)} \left[ \sum_{t=0}^{\infty} \gamma^t R(s_t, a_t) \right]. \tag{1}$$

For a given policy $\pi$, the state value function is defined as $V^\pi(s) = \mathbb{E}_\pi\left[\sum_{t=0}^\infty \gamma^t R(s_t, a_t)|s_0 = s\right]$, and the state-action value function is defined as $Q^\pi(s, a) = \mathbb{E}_\pi\left[\sum_{t=0}^\infty \gamma^t R(s_t, a_t)|s_0 = s, a_0 = a\right]$.

**Offline RL.** In offline RL, the agent is able to access a fixed dataset $\mathcal{D} = \{(s_i, a_i, r_i, s_i')\}_{i=0}^{n-1}$ collected by some behavior policy $\pi_\beta$, and aims to learn an optimal policy without further data collection [43, 44]. Standard Q learning methods seek to learn the optimal Q function by minimizing:

$$L_Q(\theta) = \mathbb{E}_{(s,a,s')\sim\mathcal{D}}\left[(Q_\theta(s,a) - R(s,a) - \gamma \max_{a'} Q_{\theta'}(s',a'))^2\right], \tag{2}$$

where $Q_\theta(s, a)$ denotes a parameterized Q function, and $Q_{\theta'}(s, a)$ represents a target Q function with parameters updated using Polyak averaging [58].

A central challenge in offline RL is the presence of out-of-distribution (OOD) actions that fall outside the support of the behavior policy. These OOD actions often lead to inaccurate Q value estimates due to extrapolation errors [23]. As a result, maximizing the estimated Q functions tends to favor OOD actions with overestimated values, resulting in significant performance degradation [44].

# 3 Adaptive Neighborhood-Constrained Q Learning for Offline RL

This section focuses on developing action-selection constraints to address the OOD issue in offline RL. First, we provide a systematic categorization of existing approaches and analyze the strengths and limitations of each category. To overcome the limitations, we propose a new neighborhood constraint, supported by theoretical analyses that elucidate its properties. Furthermore, we design an adaptive variant of this flexible constraint, achieving pointwise conservatism in practice. Finally, we develop a simple yet effective algorithm to facilitate Q learning and policy extraction under the constraint.

## 3.1 A Categorization and Analysis of Constraints in Offline RL

To address the OOD issue, offline RL algorithms impose various constraints to prevent either the learned policy (in actor-critic training) or the Bellman target (in Q learning) from selecting OOD actions. In this context, many approaches inherently align the probability density of the trained policy with that of the behavior policy, either explicitly, through divergence measures such as reverse KL [32, 83], forward KL (i.e., behavior cloning) [21, 61], and Fisher divergence [39], or implicitly, via value penalties that reduce the Q values of trained policy's actions and while increasing those of dataset actions [42, 12]. We formalize this concept as the density constraint in Definition 1.

**Definition 1** (Density constraint). *The trained policy satisfies the density constraint $\mathrm{D}(\pi, \pi_\beta) \le \epsilon$, where $\mathrm{D}$ represents a divergence measure between the trained policy $\pi$ and the behavior policy $\pi_\beta$, e.g., KL, total variation (TV), or Fisher divergence.*

While straightforward, this density constraint can be overly restrictive in many scenarios. Lemma 1 provides a theoretical upper bound on policy performance under various forms of density constraints.

**Lemma 1** (Performance bound under density constraints). *If any of the conditions $\mathrm{D}_{\mathrm{KL}}(\pi\|\pi_\beta) \le 2\epsilon$, $\mathrm{D}_{\mathrm{KL}}(\pi_\beta\|\pi) \le 2\epsilon$, or $\mathrm{D}_{\mathrm{TV}}(\pi, \pi_\beta) \le \sqrt{\epsilon}$ holds, then the policy performance $\eta$ is bounded as follows:*

$$\eta(\pi) \le \eta(\pi_\beta) + \frac{2R_{\max}}{(1-\gamma)^2}\sqrt{\epsilon}. \tag{3}$$

Lemma 1 demonstrates that policy performance under the density constraints is affected by the overall quality of the behavior policy. Consequently, even if the optimal behavior is present in the dataset, the learned policy can still remain highly suboptimal if the overall behavior policy is of low quality.

To address this limitation inherent in the density constraint, recent studies have explored the more relaxed support constraint, which only requires selected actions to be within the support of the behavior policy [82, 26, 53, 93]. Prior to the formal definition, we introduce a general loss function for constrained Q learning in Eq. (4), where $\mathcal{C}(s)$ denotes a conditional set of actions for a given state.

$$L_\mathcal{C}(\theta) = \mathbb{E}_{(s,a,s')\sim\mathcal{D}}\left[(Q_\theta(s,a) - R(s,a) - \gamma \max_{a'\in\mathcal{C}(s')} Q_{\theta'}(s',a'))^2\right]. \tag{4}$$

**Definition 2** (Support constraint). *The selected action in the Bellman target is restricted to the support of the behavior policy, which is defined as $\mathcal{C}_{\mathrm{Supp}}(s) := \{a \in \mathcal{A} \mid \pi_\beta(a|s) > \epsilon\}$, where $\pi_\beta$ is the behavior policy and $\epsilon$ is a threshold that determines the support.*

This support constraint is generally considered the least restrictive for offline RL [82], as the quality of actions outside the behavior policy's support cannot be reliably assessed. To enforce this constraint, existing approaches typically rely on behavior policy modeling, using techniques such as conditional variational autoencoders (CVAEs) [23, 41, 94, 82], autoregressive models [26], flow-GANs [93], and diffusion models [9]. Specifically, these methods either use pre-trained behavior density estimators to explicitly constrain the policy within the behavior support [41, 82, 93], or employ pre-trained behavior policy samplers to generate in-support actions and select the one with the highest Q value [23, 26, 94, 9]. However, their effectiveness is fundamentally limited by the accuracy of behavior policy modeling [26], which is well-known to be challenging due to the high-dimensional and multi-modal nature of real-world data [93]. Moreover, these methods incur extra computational costs due to behavior model training and, in some cases, extensive action sampling per state.

A parallel research direction has introduced the sample constraint, which constructs the Bellman target exclusively using actions in the dataset [40, 85, 92, 88], and derives policies via weighted behavior cloning [61, 81], thereby avoiding querying any out-of-dataset actions.

**Definition 3** (Sample constraint). *The selected action in the Bellman target is restricted to the sample set $\mathcal{C}_{\mathrm{Samp}}(s) := \{a \in \mathcal{A} \mid (s, a) \in \mathcal{D}\}$, consisting of actions in the dataset for a given state $s \in \mathcal{D}$.*

Sample constraint methods are computationally efficient, easy to implement, and effective in avoiding extrapolation errors [40]. However, their performance is inherently constrained by the inability to generalize beyond the offline dataset. This over-conservatism becomes especially problematic when the dataset lacks coverage of near-optimal actions, a common issue in environments with large or continuous action spaces, or when the dataset exhibits low quality or limited diversity. Moreover, to ensure computational stability, these methods often struggle to adequately suppress the impact of suboptimal dataset actions [88], diminishing their efficacy when such actions dominate the dataset.

For extended discussions on related work, we refer the reader to Appendix A.

## 3.2 Neighborhood Constraint for Offline RL

This work aims to mitigate the over-conservatism inherent in the density and sample constraints, while circumventing the behavior modeling requirement posed by the support constraint. To this end, we introduce a flexible neighborhood constraint for offline RL, which restricts action selection in the Bellman target to the union of neighborhoods of dataset actions on a given state.

**Definition 4** (Neighborhood constraint). *The selected action in the Bellman target is restricted to the neighborhood set $\mathcal{C}_{\mathrm{N}}(s) := \{\tilde{a} \in \mathcal{A} \mid \|\tilde{a} - a\| \le \epsilon, (s, a) \in \mathcal{D}\}$, which comprises actions located within the $\epsilon$-neighborhoods of all dataset actions on a given state $s \in \mathcal{D}$.*

In contrast to the sample constraint, this neighborhood constraint offers greater freedom, as it allows for seeking better actions beyond the dataset, within a flexible range. As shown in the following Theorem 1, the neighborhood constraint can serve as a viable approximation to the least restrictive support constraint, with the benefit of being achievable without behavior policy modeling. To establish the theorem, we introduce the standardness assumption commonly used in geometric measure theory [13, 8], which ensures that the measure does not exhibit "holes" at small scales.

**Assumption 1** (Standardness). *Let $S \subseteq \mathbb{R}^d$ be the support of a probability distribution $\nu$, and $B(x, r)$ be the closed ball of radius $r$ centered at $x \in \mathbb{R}^d$. There exist constants $r_0 > 0$ and $C_0 > 0$ such that:*

$$\forall x \in S, \ \forall r \le r_0, \ \nu(B(x, r)) \ge C_0 \cdot r^d. \tag{5}$$

**Theorem 1** (Support approximation via neighborhoods). *Let $S \subseteq \mathbb{R}^d$ be the compact support of a distribution $\nu$, and let $X_1, \dots, X_n$ be independent and identically distributed samples from $\nu$. Define $U_{n,\epsilon} = \bigcup_{i=1}^n B(X_i, \epsilon)$ as the union of closed balls of radius $\epsilon$ centered at the samples. Let $\mathcal{N}(S, \epsilon/2)$ denote the covering number of $S$, i.e., the minimal number of $\epsilon/2$-balls required to cover $S$. Under the standardness Assumption 1 with constants $r_0, C_0 > 0$, for any $\delta \in (0, 1)$ and $\epsilon \le 2r_0$, if*

$$n \ge \frac{1}{C_0(\epsilon/2)^d} \left( \log \mathcal{N}(S, \epsilon/2) + \log(1/\delta) \right), \tag{6}$$

*then with probability at least $1 - \delta$, the Hausdorff distance between $S$ and $U_{n,\epsilon}$ satisfies*

$$d_H(S, U_{n,\epsilon}) := \max \left( \sup_{x \in S} \inf_{u \in U_{n,\epsilon}} d(x, u), \sup_{u \in U_{n,\epsilon}} \inf_{x \in S} d(x, u) \right) \leq \epsilon. \tag{7}$$

Since both the support and neighborhood constraint sets are defined over actions conditioned on a given state, Theorem 1 analyzes their relationship at a fixed state, focusing on their difference in the action space. In Theorem 1, $\nu$ represents the behavior policy distribution at a state, and $S$ is defined as its support. $X_1, \ldots, X_n$ are i.i.d. samples from $\nu$, corresponding to dataset actions at that state. This theorem ensures that the union of sample neighborhoods $U_{n,\epsilon}$ approximates the support $S$ within a controlled Hausdorff distance, capturing the trade-off between sample size $n$, neighborhood radius $\epsilon$, and the geometric complexity of support $S$ (as reflected by $\mathcal{N}(S, \epsilon/2)$). Note that this Hausdorff distance is sensitive to outliers [29], making it well-suited for evaluating approximation quality in our setting, where outlier actions can significantly affect Q function optimization.

In the following, we further investigate several properties of the proposed neighborhood constraint in the context of controlling extrapolation and distribution shift. The definition of this constraint is closely related to the concept of extrapolation, and Lemma 2 provides a theoretical characterization of the extrapolation behavior of deep Q functions under this constraint.

**Lemma 2** (Extrapolation behavior). *Under the neural tangent kernel (NTK) regime [30], for any in-sample state-action pair $(s, a) \in \mathcal{D}$ and in-neighborhood state-action pair $(s, \tilde{a})$ such that $\|\tilde{a} - a\| \leq \epsilon$, the value difference of the deep Q function can be bounded as:*

$$\|Q_\theta(s, \tilde{a}) - Q_\theta(s, a)\| \leq C(\sqrt{\min(\|s \oplus a\|, \|s \oplus \tilde{a}\|)}\sqrt{\epsilon} + 2\epsilon), \tag{8}$$

*where $\oplus$ denotes the vector concatenation operation, and $C$ is a finite constant.*

Lemma 2 is a direct corollary of Theorem 1 in [45], specialized to the case of action extrapolation. It demonstrates that, for any unseen action $\tilde{a}$, its Q value $Q_\theta(s, \tilde{a})$ can be effectively controlled by a dataset action's Q value $Q_\theta(s, a)$ and the distance $\|\tilde{a} - a\|$. Specifically, a smaller neighborhood radius yields tighter control over the output of deep Q functions.

Furthermore, Proposition 1 demonstrates that, under mild continuity conditions on the transition dynamics [15, 87], the neighborhood constraint also helps to bound the degree of distribution shift.

**Proposition 1** (Distribution shift). *Let $\pi_1$ be a deterministic policy that satisfies the neighborhood constraint with threshold $\epsilon$. Assume that the transition dynamics $P$ is $K_P$-Lipschitz continuous: $\forall s \in \mathcal{S}, \forall a_1, a_2 \in \mathcal{A}, \|P(s'|s, a_1) - P(s'|s, a_2)\| \leq K_P\|a_1 - a_2\|$. Then, there exists a policy $\pi_2$ satisfying the sample constraint such that:*

$$\mathrm{D}_{\mathrm{TV}}\left(d^{\pi_1}(\cdot), d^{\pi_2}(\cdot)\right) \leq \frac{\gamma K_P \epsilon}{2(1-\gamma)}, \tag{9}$$

*where $d^\pi(s) = (1 - \gamma) \sum_{t=0}^\infty \gamma^t \mathbb{E}_\pi \left[\mathbb{I}\left[s_t = s\right]\right]$ is the state occupancy induced by policy $\pi$.*

**Adaptive neighborhoods.**   The neighborhood constraint is highly flexible and can, in practice, achieve pointwise conservatism by adapting the neighborhood radius for each data point, enabling the design of an adaptive neighborhood constraint. Considering real-world data patterns, expert data typically clusters within a narrow distribution [3, 18], necessitating tighter constraints to mitigate extrapolation errors, while suboptimal data tends to be more dispersed and thus benefits from looser constraints that facilitate policy improvement. Inspired by this idea, we propose a concrete instantiation of adaptive neighborhoods in Definition 5, where per-sample radius is set as $\epsilon \exp(-\alpha A(s, a))$.

**Definition 5** (Adaptive neighborhood constraint). *The selected action in the Bellman target is restricted to the adaptive neighborhood set $\mathcal{C}_{\mathrm{AN}}(s) := \{\tilde{a} \in \mathcal{A} \mid \|\tilde{a} - a\| \leq \epsilon \exp(-\alpha A(s, a)), (s, a) \in \mathcal{D}\}$, where $A$ denotes the advantage function and $\alpha$ is an inverse temperature parameter that modulates the sensitivity of the neighborhood radius to advantage values.*

This adaptive neighborhood constraint assigns larger neighborhood radii to dataset actions with low advantage, thereby promoting a broader search over the action space and further mitigating the impact of low-quality data. Conversely, dataset actions with high advantage are assigned smaller neighborhood radii to more effectively reduce the overall extrapolation error. In practice, advantage

estimation errors are typically not a concern for two reasons: (1) In-distribution estimation: the advantage is computed only on dataset points $(s, a) \in \mathcal{D}$, where estimates are relatively reliable; (2) Qualitative use: the purpose of using advantage is to distinguish actions qualitatively, and the exponential form is merely a soft heuristic to bias the radius, without requiring precise values.

### 3.3 Adaptive Neighborhood-Constrained Q Learning

In the following, we develop an efficient bilevel optimization framework to achieve Q learning under the adaptive neighborhood constraint. Specifically, we aim to minimize the following Q learning loss:

$$L_{\text{ANQ}}(\theta) = \mathbb{E}_{(s,a,s')\sim\mathcal{D}} \left[ \left( Q_\theta(s,a) - R(s,a) - \gamma \max_{a'\in\mathcal{C}_{\text{AN}}(s')} Q_{\theta'}(s',a') \right)^2 \right]. \quad (10)$$

**Bilevel optimization.**  The primary challenge in constrained Q learning lies in enforcing $\max_{a'\in\mathcal{C}(s')}$ in the Bellman target. While the support constraint $\mathcal{C}_{\text{Supp}}$ typically necessitates accurate modeling of the behavior policy, we demonstrate that the adaptive neighborhood constraint $\mathcal{C}_{\text{AN}}$ can be effectively enforced by decomposing the objective into a bilevel optimization structure:

$$\max_{a\in\mathcal{C}_{\text{AN}}(s)} Q(s,a), \ \forall s \in \mathcal{D} \iff \begin{array}{l} \displaystyle\max_{a\in\mathcal{D}(s)} Q(s, a + \delta_{sa}), \ \forall s \in \mathcal{D} \\ \text{s.t. } \delta_{sa} = \displaystyle\operatorname*{argmax}_{\|\delta\| \leq \epsilon\exp(-\alpha A(s,a))} Q(s, a + \delta), \ \forall (s,a) \in \mathcal{D}, \end{array} \quad (11)$$

where we use $\mathcal{D}(s)$ to denote the empirical action set observed in the dataset for a given state $s \in \mathcal{D}$.

**The inner maximization** in Eq. (11) optimizes the Q function separately within each dataset action's neighborhood. To this end, we introduce an auxiliary policy $\mu_\omega$ that takes state-action pairs $(s, a)$ from the dataset as input and outputs action variations $\delta$. This formulation enables straightforward enforcement of the adaptive neighborhood constraint by restricting the norm of $\mu_\omega(s, a)$ to stay within the bound $\epsilon\exp(-\alpha A(s, a))$. Practically, we multiply both sides of the constraint inequality by $\exp(\alpha A(s, a))$ to maintain a constant constraint threshold: $\exp(\alpha A(s, a))\|\mu_\omega(s, a)\| \leq \epsilon$. Consequently, we optimize the Q function with respect to $\mu_\omega$ to seek the optimal action within the adaptive neighborhood of each dataset action, according to the following objective:

$$\max_{\mu_\omega} Q_\theta(s, a + \mu_\omega(s,a)) \ \text{ s.t. } \exp(\alpha A(s,a))\|\mu_\omega(s,a)\| \leq \epsilon, \ \forall (s,a) \in \mathcal{D}. \quad (12)$$

We reformulate the constrained optimization problem into an unconstrained one using a Lagrange multiplier $\lambda \in \mathbb{R}^+$. In addition, we introduce a state value function $V_\psi(s)$, whose training objective will be specified later, and use the difference $Q_{\theta'} - V_\psi$ to compute the advantage function. Accordingly, we optimize the following objective for the inner maximization:

$$\max_{\mu_\omega} \mathbb{E}_{(s,a)\sim\mathcal{D}} \left[ Q_\theta(s, a + \mu_\omega(s,a)) - \lambda \exp(\alpha(Q_{\theta'}(s,a) - V_\psi(s)))\|\mu_\omega(s,a)\| \right]. \quad (13)$$

**The outer maximization** in Eq. (11) searches over all dataset actions on a given state and seeks the one whose corresponding neighborhood yields the highest Q value. To achieve this objective, we first sample state-action pairs from the dataset and refine the actions by adding the outputs of the trained auxiliary policy, thereby simulating the sampling of the optimized actions across the neighborhoods. We then employ expectile regression [40] to implicitly maximize the Q function over these optimized actions. Specifically, we fit a $V$ function with the following asymmetric squared error loss, treating the Q values of the optimized actions as regression targets:

$$\min_{V_\psi} \mathbb{E}_{(s,a)\sim\mathcal{D}} \left[ L_2^\tau \left( Q_{\theta'}(s, a + \mu_{\omega'}(s,a)) - V_\psi(s) \right) \right], \quad (14)$$

where $L_2^\tau(x) = |\tau - \mathbb{1}(x < 0)|x^2$, $\tau \in (0,1)$, $Q_{\theta'}$ and $\mu_{\omega'}$ are the target $Q$ function and target auxiliary policy, whose parameters are updated via Polyak averaging [58].

For $\tau \approx 1$, $V_\psi(s)$ captures the maximum Q value within the adaptive neighborhood set $\mathcal{C}_{\text{AN}}(s)$. By substituting $\max_{a'\in\mathcal{C}_{\text{AN}}(s')} Q_{\theta'}(s', a')$ in Eq. (10) with $V_\psi(s')$, adaptive neighborhood-constrained Q learning is achieved based on the following loss:

$$\min_{Q_\theta} \mathbb{E}_{(s,a,s')\sim\mathcal{D}} \left[ \left( Q_\theta(s,a) - R(s,a) - \gamma V_\psi(s') \right)^2 \right]. \quad (15)$$

**A radius-agnostic framework.** Although our Q learning algorithm is presented specifically for the adaptive neighborhood in Definition 5, i.e., using the per-sample radius $\epsilon \exp(-\alpha A(s, a))$, the overall framework is general and can accommodate arbitrary radius schemes. Specifically, one can simply replace $\epsilon \exp(-\alpha A(s, a))$ in Definition 5 (and Eq. (11)) with $\epsilon f(s, a)$ to define a generic per-sample neighborhood radius, where $f : \mathcal{S} \times \mathcal{A} \to \mathbb{R}^+$ is an arbitrary function that modulates the radius. Correspondingly, in Eq. (12), $\exp(\alpha A(s, a))$ becomes $1/f(s, a)$, and Eq. (13) becomes:

$$\max_{\mu_\omega} \mathbb{E}_{(s,a) \sim \mathcal{D}}[Q_\theta(s, a + \mu_\omega(s, a)) - \lambda \|\mu_\omega(s, a)\|/f(s, a)]. \tag{16}$$

With all other equations unchanged, the resulting algorithm supports arbitrary neighborhood schemes.

### 3.4 Policy Extraction via Weighted Regression Toward Optimized Actions

While our algorithm enables Q learning under the adaptive neighborhood constraint, it does not explicitly derive the corresponding policy, thereby requiring a separate policy extraction step. To this end, we employ the weighed behavior cloning method [17, 11] and, rather than imitating the actions in the dataset, we instead imitate the actions that have been refined through the auxiliary policy, which represents the optimal actions within the adaptive neighborhoods. Moreover, we set the weights as the exponentiated advantage function [79, 61, 60, 81]. Consequently, the final policy $\pi_\phi : \mathcal{S} \to \mathcal{A}$ is extracted according to the following loss:

$$\min_{\pi_\phi} \mathbb{E}_{(s,a) \sim \mathcal{D}} \exp(\beta(Q_{\theta'}(s, a + \mu_\omega(s, a)) - V_\psi(s)))\|a + \mu_\omega(s, a) - \pi_\phi(s)\|_2^2, \tag{17}$$

where $\beta$ is an inverse temperature and $Q_{\theta'} - V_\psi$ computes the advantage function.

**Remark.** This policy extraction step, which does not interfere with the Q learning process described in Section 3.3, also constitutes a key distinction from existing regression-based policy learning objectives, such as those employed in AWR [61], AWAC [60], CRR [81], 10% BC [10], IQL [40], and SQL [88], all of which perform weighted regression toward the dataset actions. In contrast, our policy learning objective performs weighted regression toward the optimized actions within the adaptive neighborhoods. This enables the trained policy to select actions superior to those in the dataset, while also significantly mitigating the adverse effects of suboptimal dataset actions.

---
**Algorithm 1** ANQ
---
1: Initialize policy $\pi_\phi$, auxiliary policy $\mu_\omega$, target auxiliary policy $\mu_{\omega'}$, Q-network $Q_\theta$, target Q-network $Q_{\theta'}$, and V-network $V_\psi$.
2: **for** each gradient step **do**
3:     Update $\psi$ by minimizing Eq. (14)
4:     Update $\theta$ by minimizing Eq. (15)
5:     Update $\omega$ by maximizing Eq. (13)
6:     Update $\phi$ by maximizing Eq. (17)
7:     Update target networks: $\theta' \leftarrow (1 - \xi)\theta' + \xi\theta$, $\omega' \leftarrow (1 - \xi)\omega' + \xi\omega$
8: **end for**
---

Integrating all components, we present our final algorithm in Algorithm 1.

## 4 Experiments

We conduct experiments to evaluate the performance and properties of the proposed approach ANQ. Experimental details and extended results are provided in Appendices C and D, respectively.

### 4.1 Benchmark Results

**Tasks.** We assess ANQ on two distinct task suites from D4RL [20]: the Gym-MuJoCo locomotion domains and the challenging AntMaze domains. The AntMaze tasks involve sparse rewards and require the ant agent to combine segments of suboptimal trajectories to reach the maze's goal.

**Baselines.** Our offline RL baselines span various constraint categories. For density constraints, we compare to TD3BC [21], CQL [42], and AWAC [60], where CQL and AWAC essentially enforce a density constraint as analyzed in Theorem 3.5 of [42] and Section 3 of [53], respectively. For support constraints, we include BCQ [23], BEAR [41], and SPOT [82]. For sample constraints, we compare against OneStep RL [6] and IQL [40]. We also include the sequence-modeling approach DT [10].

Table 2: Averaged normalized scores on Gym locomotion and Antmaze tasks over five random seeds. m = medium, m-r = medium-replay, m-e = medium-expert, e = expert, r = random; u = umaze, u-d = umaze-diverse, m-p = medium-play, m-d = medium-diverse, l-p= large-play, l-d = large-diverse.

| Dataset-v2 | BCQ | BEAR | DT | AWAC | OneStep | TD3BC | CQL | IQL | SPOT | ANQ (Ours) |
|---|---|---|---|---|---|---|---|---|---|---|
| halfcheetah-m | 46.6 | 43.0 | 42.6 | 47.9 | 50.4 | 48.3 | 47.0 | 47.4 | 58.4 | **61.8±1.4** |
| hopper-m | 59.4 | 51.8 | 67.6 | 59.8 | 87.5 | 59.3 | 53.0 | 66.2 | 86.0 | **100.9±0.6** |
| walker2d-m | 71.8 | -0.2 | 74.0 | 83.1 | 84.8 | 83.7 | 73.3 | 78.3 | **86.4** | 82.9±1.5 |
| halfcheetah-m-r | 42.2 | 36.3 | 36.6 | 44.8 | 42.7 | 44.6 | 45.5 | 44.2 | 52.2 | **55.5±1.4** |
| hopper-m-r | 60.9 | 52.2 | 82.7 | 69.8 | 98.5 | 60.9 | 88.7 | 94.7 | 100.2 | **101.5±2.7** |
| walker2d-m-r | 57.0 | 7.0 | 66.6 | 78.1 | 61.7 | 81.8 | 81.8 | 73.8 | **91.6** | 92.7±3.8 |
| halfcheetah-m-e | **95.4** | 46.0 | 86.8 | 64.9 | 75.1 | 90.7 | 75.6 | 86.7 | 86.9 | 94.2±0.8 |
| hopper-m-e | **106.9** | 50.6 | **107.6** | 100.1 | **108.6** | 98.0 | 105.6 | 91.5 | 99.3 | 107.0±4.9 |
| walker2d-m-e | 107.7 | 22.1 | 108.1 | 110.0 | **111.3** | 110.1 | 107.9 | 109.6 | **112.0** | 111.7±0.2 |
| halfcheetah-e | 89.9 | 92.7 | 87.7 | 81.7 | 88.2 | **96.7** | 96.3 | 95.0 | 94.8 | 95.9±0.4 |
| hopper-e | 109.0 | 54.6 | 94.2 | 109.5 | 106.9 | 107.8 | 96.5 | 109.4 | **111.0** | 111.4±2.5 |
| walker2d-e | 106.3 | 106.6 | 108.3 | **110.1** | **110.7** | 110.2 | 108.5 | 109.9 | 109.9 | 111.8±0.1 |
| halfcheetah-r | 2.2 | 2.3 | 2.2 | 6.1 | 2.3 | 11.0 | 17.5 | 13.1 | **25.4** | 24.9±1.0 |
| hopper-r | 7.8 | 3.9 | 5.4 | 9.2 | 5.6 | 8.5 | 7.9 | 7.9 | 23.4 | **31.1±0.2** |
| walker2d-r | 4.9 | **12.8** | 2.2 | 0.2 | 6.9 | 1.6 | 5.1 | 5.4 | 2.4 | 11.2±9.5 |
| locomotion total | 968.0 | 581.7 | 972.6 | 975.3 | 1041.2 | 1013.2 | 1010.2 | 1033.1 | 1139.9 | **1194.5** |
| antmaze-u | 78.9 | 73.0 | 54.2 | 80.0 | 54.0 | 73.0 | 82.6 | 89.6 | **93.5** | **96.0±1.6** |
| antmaze-u-d | 55.0 | 61.0 | 41.2 | 52.0 | 57.8 | 47.0 | 10.2 | 65.6 | 40.7 | **80.2±1.8** |
| antmaze-m-p | 0.0 | 0.0 | 0.0 | 0.0 | 0.0 | 0.0 | 59.0 | **76.4** | 74.7 | 76.2±3.3 |
| antmaze-m-d | 0.0 | 8.0 | 0.0 | 0.2 | 0.6 | 0.2 | 46.6 | 72.8 | **79.1** | 77.2±6.1 |
| antmaze-l-p | 6.7 | 0.0 | 0.0 | 0.0 | 0.0 | 0.0 | 16.4 | 42.0 | 35.3 | **56.2±4.9** |
| antmaze-l-d | 2.2 | 0.0 | 0.0 | 0.0 | 0.2 | 0.0 | 3.2 | 46.0 | 36.3 | **55.8±4.0** |
| antmaze total | 142.8 | 142.0 | 95.4 | 132.2 | 112.6 | 120.2 | 218.0 | 392.4 | 359.6 | **441.6** |

**Comparisons.** Aggregated results are reported in Table 2. On the Gym locomotion tasks, ANQ outperforms existing methods on most tasks and achieves the highest overall score. On the challenging AntMaze tasks, ANQ surpasses the baselines by a considerable margin, particularly in the most complex large maze settings. The learning curves are provided in Appendix D.5. We also extend our evaluation in Appendix D.2 by comparing ANQ with additional recent SOTA algorithms.

**Runtime.** We test the runtime of ANQ and some baseline methods on a GeForce RTX 3090. As shown in Appendix D.1, ANQ is among the fastest tier of offline RL algorithms, on par with efficient baselines such as AWAC, IQL, and TD3BC, with a detailed analysis provided in the same section.

## 4.2 Noisy Data Results

We examine the robustness of various constraint types under noisy data conditions. Specifically, we construct noisy datasets by mixing the random and expert datasets at varying ratios, thereby simulating real-world scenarios such as imperfect demonstrations in robotics or suboptimal data collection in autonomous systems. We then evaluate the performance of representative algorithms, including CQL (density), IQL (sample), SPOT (support), and ANQ (neighborhood).

As presented in Figure 1(a), ANQ generally outperforms the other algorithms across expert ratios, and its performance advantage becomes more pronounced as the proportion of expert data decreases. As analyzed in Section 3.1, the density constraint is sensitive to the overall quality of the behavior policy, which tends to be low in noisy datasets, while the support constraint often struggles with modeling the multi-modal behavior policy distribution inherent in such datasets. In contrast, the adaptive neighborhood constraint employed by ANQ exhibits greater robustness to noisy data. Moreover, we evaluate ANQ on such noisy datasets with varying inverse temperature $\alpha$ that controls the adaptiveness of neighborhood radius. The results in Figure 1(b) demonstrate that, compared with the uniform neighborhood constraint ($\alpha = 0$), the adaptive neighborhood constraint further mitigates the adverse effects of low-quality data in the datasets and exhibits greater robustness to such data.

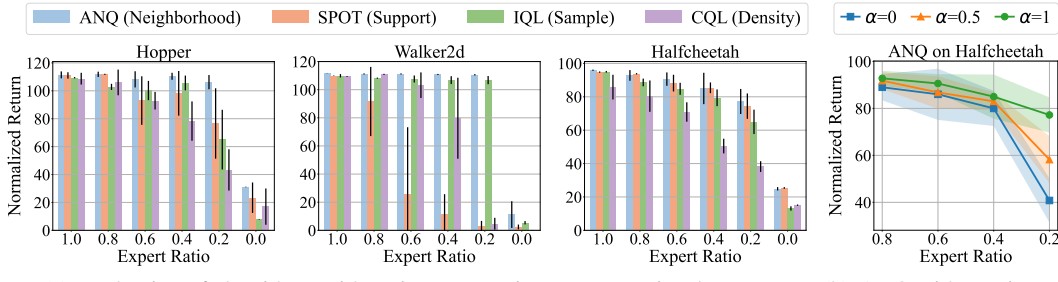

(a) Evaluation of algorithms with various constraint types on noisy datasets

(b) ANQ with varying $\alpha$

Figure 1: (a) Evaluation on noisy datasets over five random seeds. (b) Evaluation of ANQ on noisy datasets with varying inverse temperature $\alpha$ that determines the adaptiveness of neighborhood radius.

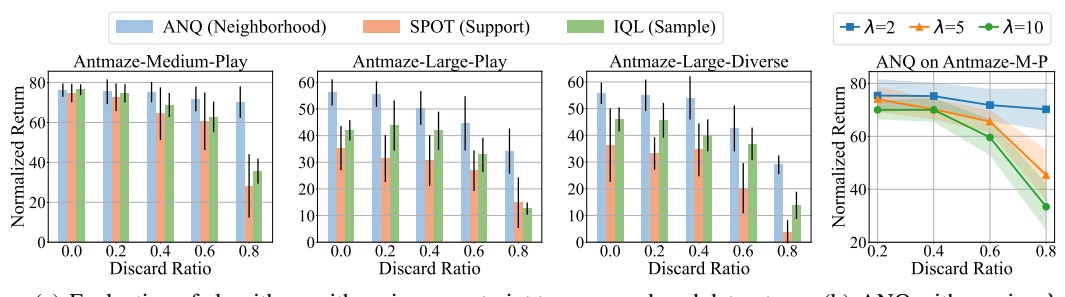

(a) Evaluation of algorithms with various constraint types on reduced datasets

(b) ANQ with varying $\lambda$

Figure 2: (a) Evaluation on reduced datasets over five random seeds. (b) Evaluation of ANQ on reduced datasets with varying Lagrange multiplier $\lambda$ that controls the overall radius of neighborhoods.

### 4.3 Limited Data Results

We also investigate the robustness of these constraint types in limited data settings. To this end, we create reduced datasets by randomly discarding some portion of transitions from the AntMaze datasets. This setup mimics practical scenarios in which data is rare or partially missing, such as in healthcare applications. Again, we evaluate the performance of IQL, SPOT, and ANQ, omitting CQL due to its consistently inferior performance on Antmaze tasks as reported in Table 2.

As shown in Figure 2(a), ANQ demonstrates superior performance across nearly all discard ratios, with the performance gap widening as the amount of available data decreases. In such limited data settings, support constraint methods face even more difficulties in modeling the behavior policy due to sample scarcity, whereas sample constraint methods risk being overly conservative because of reduced coverage of near-optimal actions. In contrast, ANQ bypasses the need to model the behavior policy and leverages generalization to attain superior performance beyond the offline dataset. Furthermore, we evaluate ANQ on reduced datasets with varying Lagrange multiplier $\lambda$, which is inversely proportional to the overall radius of the neighborhoods. The results in Figure 1(b) show that an appropriately large neighborhood is crucial for achieving good performance in such limited data scenarios, further showcasing the benefit and flexibility of ANQ over sample constraint methods.

### 4.4 Ablation Study

**Lagrange multiplier $\lambda$.** The Lagrange multiplier $\lambda$ controls the overall neighborhood radius in ANQ. We vary $\lambda$ and present the learned Q values and performance across various tasks in Figure 3. As $\lambda$ decreases from a sufficiently large value, ANQ enables larger neighborhoods, resulting in higher and possibly divergent Q values, and performance also exhibits a rise-then-fall trend. Note that ANQ with $\lambda = 0$ and $\lambda = \infty$ approximately corresponds to Q learning without any constraint and with the sample constraint, respectively. Therefore, the results provide evidence that ANQ not only effectively suppresses extrapolation error, but also mitigates the over-conservatism of the sample constraint.

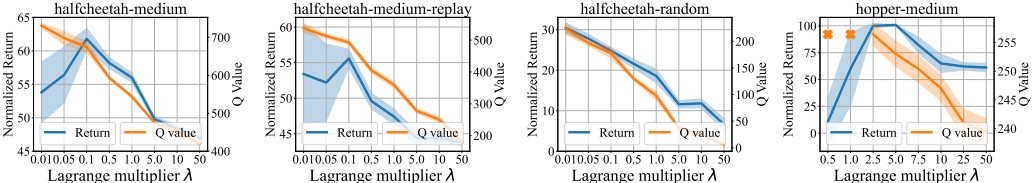

Figure 3: Performance and Q values of ANQ with varying Lagrange multiplier $\lambda$ over five random seeds. The crosses $\times$ mean that the value functions diverge in some seeds. As $\lambda$ decreases, ANQ enables larger overall neighborhood radii, resulting in higher and probably divergent learned Q values. A moderate $\lambda$ (neighborhood constraint) is crucial for achieving superior performance.

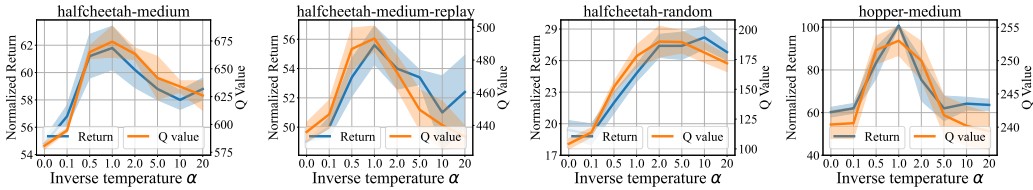

Figure 4: Performance and Q values of ANQ with varying inverse temperature $\alpha$ over five random seeds. An appropriately large $\alpha$ (adaptive neighborhoods) yields enhanced performance.

**Inverse temperature $\alpha$.** The inverse temperature $\alpha$ determines how the neighborhood radius adapts to the action advantage, where $\alpha = 0$ corresponds to a fixed neighborhood radius. The results in Figure 4 demonstrate that an appropriately large $\alpha$ leads to enhanced performance, validating our design of advantage-based adaptive neighborhoods. However, an excessively large $\alpha$ ($\alpha = 20$) may degrade performance, likely due to the increased variance of the learning objective.

## 5 Conclusion and Limitations

This work focuses on developing action-selection constraints to address the OOD issue in offline RL. To overcome the identified limitations of existing approaches, we propose the flexible neighborhood constraint and the corresponding algorithm ANQ, which mitigates the over-conservatism inherent in the density and sample constraints, and approximates the least restrictive support constraint without challenging behavior modeling. Empirical results demonstrate that ANQ achieves SOTA performance on standard offline RL benchmarks and exhibits enhanced robustness to noisy or limited data.

At the algorithmic level, this work develops a general framework for achieving pointwise conservatism by adapting the neighborhood radius for each data point. In particular, the practical algorithm ANQ represents one instantiation of adaptive neighborhoods, using data point quality as the adaptation criterion. However, this design is not necessarily optimal; incorporating additional information, such as uncertainty quantification, could potentially lead to more effective neighborhood construction.

## Acknowledgment

We thank the anonymous reviewers for feedback on an early version of this paper. This work was supported by the National Key R&D Program of China under Grant 2018AAA0102801.

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

# A  Extended Related Work

**Model-free offline RL.**   Offline RL aims to learn policies from a fixed dataset without any further interaction with the environment [43, 44]. Standard off-policy methods often struggle in this setting due to extrapolation errors induced by OOD actions [23]. To address this issue, a variety of algorithms have been developed, broadly falling into model-free and model-based categories. Within the model-free family, value regularization methods promote conservative value estimation, either by explicitly penalizing overestimated Q values [42, 39, 48, 86, 12, 69, 54], or by employing ensembles to capture epistemic uncertainty [2, 5, 89]. Policy constraint approaches, by contrast, aim to keep the learned policy close to the behavior policy, accomplished either explicitly through divergence penalties [83, 41, 32, 21, 82], implicitly via weighted behavior cloning [11, 61, 60, 81, 53], or through carefully designed policy parameterizations [23, 26, 94]. More recently, these methods have increasingly shifted toward learning the optimal policy within the support of the behavior policy, offering theoretical guarantees and reduced sensitivity to the overall quality of datasets [82, 53, 54, 50]. Another direction emphasizes in-sample learning, which avoids estimating values of unobserved actions by constructing Bellman targets solely from dataset samples [6, 47, 40, 85, 88, 25, 92]. For example, OneStep RL [6] employs SARSA-style value updates [73] and performs a single round of policy improvement. IQL [40] extends this idea using expectile regression within the SARSA update, enabling multi-step dynamic programming. In contrast to strict in-sample learning, some methods propose to exploit mild generalization beyond the dataset to further boost performance [56, 49]. Orthogonal to the above, another influential branch of work formulates offline RL as conditional sequence modeling [10]. These methods employ causal transformers conditioned on the desired return, past states, and actions to autoregressively generate future actions [10, 84, 51, 28], bypassing the need for bootstrapping in long-term credit assignment [62, 64].

**Model-based offline RL.**   Model-based offline RL approaches construct a learned model of the environment dynamics, which is subsequently used to generate synthetic trajectories for policy improvement [72, 31, 34]. Given the challenges of distributional shift in offline settings, algorithms such as MOPO [90] and MOReL [36] incorporate mechanisms to quantify the model uncertainty and penalize high-uncertainty state-action pairs, thereby discouraging the exploitation of unreliable model predictions. MOBILE [71] further advances this line of work by proposing an uncertainty quantification approach based on inconsistencies in Bellman estimations within an ensemble of learned dynamics models. Some algorithms bridge the gap between model-based and model-free paradigms by incorporating similar regularization techniques. For instance, COMBO [91] integrates value penalization into model-based rollouts, while BREMEN [57] emphasizes behavior-regularized policy optimization. More recently, SCAS [55] proposes a generic model-based regularizer for model-free offline RL that unifies OOD state correction and OOD action suppression. Despite their promise, model-based methods often entail substantial computational costs [31], and their efficacy is highly contingent on the fidelity of the learned dynamics model [59].

**Neighborhood in offline RL.**   This work formalizes the neighborhood constraint for offline RL and demonstrates its key properties through theoretical analyses. Some offline RL methods implicitly involve the concept of neighborhood by incorporating regularization terms that guide the trained policy toward dataset actions [21, 75, 66, 56]. However, due to the multi-modal nature of dataset action distributions [9] and the limited expressiveness of policy architectures [80], these policy iteration approaches cannot ensure that the trained policy's output actions for the Bellman target remain within the union of dataset action neighborhoods. In contrast, we adopt a value iteration framework and introduce an auxiliary policy to effectively enforce the proposed neighborhood constraint. BCQ [23] employs a perturbation model related to our auxiliary policy. Specifically, BCQ samples multiple candidate actions per state from a pre-trained generative behavior policy model, perturbs them using the perturbation model, and selects the one with the highest Q value. The perturbation model constructs neighborhoods around the generated actions and is intended to avoid "sampling from the generative model for a prohibitive number of times [23]". Subsequent works such as EMaQ [26] and SfBC [9] show that this perturbation model can be omitted without sacrificing performance. By contrast, our auxiliary policy constructs neighborhoods directly around dataset actions and is designed to support Q learning under the proposed constraint, without requiring complex behavior policy modeling. Based on the proposed neighborhood constraint, this work provides a general constrained Q learning framework for adaptive distribution-shift control, with the potential to extend to broader RL paradigms such as safe RL [1, 27, 24] and meta RL [78, 77, 65].

# B  Proofs

Lemma 1 in the main text characterizes the performance difference between two policies subject to divergence constraints, which is related to the results in TRPO [68] and CPO [1]. To prove Lemma 1, we begin with the performance difference lemma [35], which decomposes the difference in policy performance, $\eta(\pi') - \eta(\pi)$, into an expectation of advantages.

**Lemma 3.** *For any two policies $\pi'$ and $\pi$, the following equality holds:*

$$\eta(\pi') - \eta(\pi) = \frac{1}{1 - \gamma} \sum_s d_{\pi'}(s) \sum_a [\pi'(a|s) A^\pi(s,a)] \tag{18}$$

*where $d^\pi(s) = (1 - \gamma) \sum_{t=0}^{\infty} \gamma^t \mathbb{E}_\pi [\mathbb{I}[s_t = s]]$ is the state occupancy induced by the policy $\pi$.*

*Proof.* Please refer to the proof of Lemma 6.1 in Kakade and Langford [35]. □

**Lemma 4** (Performance bound under density constraint, Lemma 1). *If any of the conditions $D_{KL}(\pi \| \pi_\beta) \leq 2\epsilon$, $D_{KL}(\pi_\beta \| \pi) \leq 2\epsilon$, or $D_{TV}(\pi, \pi_\beta) \leq \sqrt{\epsilon}$ holds, then the policy performance $\eta$ is bounded as follows:*

$$\eta(\pi) \leq \eta(\pi_\beta) + \frac{2R_{\max}}{(1-\gamma)^2} \sqrt{\epsilon}. \tag{19}$$

*Proof.* By Lemma 3, we have

$$|\eta(\pi_\beta) - \eta(\pi)| = \frac{1}{1-\gamma} \left| \sum_s d_{\pi_\beta}(s) \sum_a [\pi_\beta(a|s) A^\pi(s,a)] \right| \tag{20}$$

$$= \frac{1}{1-\gamma} \left| \sum_s d_{\pi_\beta}(s) \sum_a \pi_\beta(a|s) (Q^\pi(s,a) - V^\pi(s)) \right| \tag{21}$$

$$= \frac{1}{1-\gamma} \left| \sum_s d_{\pi_\beta}(s) \left( \sum_a \pi_\beta(a|s) Q^\pi(s,a) - V^\pi(s) \right) \right| \tag{22}$$

$$= \frac{1}{1-\gamma} \left| \sum_s d_{\pi_\beta}(s) \left( \sum_a \pi_\beta(a|s) Q^\pi(s,a) - \sum_a \pi(a|s) Q^\pi(s,a) \right) \right| \tag{23}$$

$$= \frac{1}{1-\gamma} \left| \sum_s d_{\pi_\beta}(s) \sum_a (\pi_\beta(a|s) - \pi(a|s)) Q^\pi(s,a) \right| \tag{24}$$

$$\leq \frac{1}{1-\gamma} \sum_s d_{\pi_\beta}(s) \sum_a |\pi_\beta(a|s) - \pi(a|s)| |Q^\pi(s,a)| \tag{25}$$

$$\leq \frac{R_{\max}}{(1-\gamma)^2} \sum_s d_{\pi_\beta}(s) \sum_a |\pi_\beta(a|s) - \pi(a|s)| \tag{26}$$

$$= \frac{2R_{\max}}{(1-\gamma)^2} \sum_s d_{\pi_\beta}(s) D_{TV}(\pi, \pi_\beta)[s] \tag{27}$$

$$\leq \frac{\sqrt{2} R_{\max}}{(1-\gamma)^2} \sum_s d_{\pi_\beta}(s) \min \left\{ \sqrt{D_{KL}(\pi \| \pi_\beta)[s]}, \sqrt{D_{KL}(\pi_\beta \| \pi)[s]} \right\} \tag{28}$$

where the last inequality holds by Pinsker's inequality.

By substituting the TV condition, $D_{TV}(\pi, \pi_\beta) \leq \sqrt{\epsilon}$, into Eq. (27), or the KL condition, $D_{KL}(\pi \| \pi_\beta) \leq 2\epsilon$ or $D_{KL}(\pi_\beta \| \pi) \leq 2\epsilon$, into Eq. (28), we derive the final performance bound:

$$|\eta(\pi) - \eta(\pi_\beta)| \leq \frac{2R_{\max}}{(1-\gamma)^2} \sqrt{\epsilon}. \tag{29}$$

□

Lemma 2 in the main text analyzes the impact of neighborhood constraint on deep Q functions under the neural tangent kernel (NTK) regime [30]. The result is a direct corollary of Theorem 1 in [45], specialized to the case of action extrapolation. The NTK assumption is presented in Assumption 2.

**Assumption 2** (NTK assumption). *The Q function approximators are two-layer fully-connected ReLU neural networks with infinite width and are trained with an infinitesimally small learning rate.*

While not applicable to some practically advanced architectures [95], NTK remains one of the most influential theoretical frameworks for analyzing the generalization of deep neural networks. Assumption 2 is common in previous analyses on the generalization of deep neural networks [30, 4] and the convergence of deep RL [7, 46, 19].

**Lemma 5** (Extrapolation behavior, Lemma 2). *Under the neural tangent kernel (NTK) regime [30], for any in-sample state-action pair $(s, a) \in \mathcal{D}$ and in-neighborhood state-action pair $(s, \tilde{a})$ such that $\|\tilde{a} - a\| \leq \epsilon$, the value difference of the deep Q function can be bounded as:*

$$\|Q_\theta(s, \tilde{a}) - Q_\theta(s, a)\| \leq C(\sqrt{\min(\|s \oplus a\|, \|s \oplus \tilde{a}\|)}\sqrt{\epsilon} + 2\epsilon), \tag{30}$$

*where $\oplus$ denotes the vector concatenation operation, and $C$ is a finite constant.*

*Proof.* The lemma follows directly from Theorem 1 or Lemma 4 in [45]. Please refer to [45] for detailed proofs. □

Proposition 1 in the main text assumes that the transition dynamics is $K_P$-Lipschitz continuous, which is a common assumption in the theoretical RL studies [15, 16, 87, 66]. A small $K_P$ is also empirically supported by observations in physical systems, where small action changes often lead to limited next-state variation [76].

**Proposition 2** (Distribution shift, Proposition 1). *Let $\pi_1$ be a deterministic policy that satisfies the neighborhood constraint with threshold $\epsilon$. Assume that the transition dynamics $P$ is $K_P$-Lipschitz continuous: $\forall s \in \mathcal{S}, \forall a_1, a_2 \in \mathcal{A}, \|P(s'|s, a_1) - P(s'|s, a_2)\| \leq K_P\|a_1 - a_2\|$. Then, there exists a policy $\pi_2$ satisfying the sample constraint such that:*

$$\mathrm{D_{TV}}\left(d^{\pi_1}(\cdot), d^{\pi_2}(\cdot)\right) \leq \frac{\gamma K_P \epsilon}{2(1 - \gamma)}, \tag{31}$$

*where $d^\pi(s) = (1 - \gamma) \sum_{t=0}^\infty \gamma^t \mathbb{E}_\pi \left[\mathbb{I}\left[s_t = s\right]\right]$ is the state occupancy induced by policy $\pi$.*

*Proof.* We analyze the difference in state occupancy measures induced by two deterministic policies $\pi_1$ and $\pi_2$, leveraging the Lipschitz continuity of the transition dynamics.

The state occupancy measure $d^\pi$ satisfies the Bellman-flow constraints.:

$$d^\pi(s) = (1 - \gamma)d_0(s) + \gamma \sum_{s'} P(s \mid s', \pi(s'))d^\pi(s'), \tag{32}$$

where $d_0(s)$ is the initial state distribution.

For policies $\pi_1$ and $\pi_2$, their corresponding occupancy measures $d^{\pi_1}$ and $d^{\pi_2}$ satisfy:

$$d^{\pi_1}(s) - d^{\pi_2}(s) = \gamma \sum_{s'} \left[P(s \mid s', \pi_1(s'))d^{\pi_1}(s') - P(s \mid s', \pi_2(s'))d^{\pi_2}(s')\right]. \tag{33}$$

We decompose the difference into two terms:

$$d^{\pi_1}(s) - d^{\pi_2}(s) = \gamma \sum_{s'} P(s \mid s', \pi_1(s')) \left(d^{\pi_1}(s') - d^{\pi_2}(s')\right)$$
$$+ \gamma \sum_{s'} \left(P(s \mid s', \pi_1(s')) - P(s \mid s', \pi_2(s'))\right) d^{\pi_2}(s').$$

Let $\Delta(s) = |d^{\pi_1}(s) - d^{\pi_2}(s)|$ and apply the triangle inequality:

$$\Delta(s) \leq \gamma \left|\sum_{s'} P(s \mid s', \pi_1(s'))\Delta(s')\right| + \gamma \left|\sum_{s'} \left(P(s \mid s', \pi_1(s')) - P(s \mid s', \pi_2(s'))\right) d^{\pi_2}(s')\right|$$
$$\leq \gamma \sum_{s'} P(s \mid s', \pi_1(s'))\Delta(s') + \gamma \sum_{s'} |P(s \mid s', \pi_1(s')) - P(s \mid s', \pi_2(s'))| d^{\pi_2}(s'). \tag{34}$$

Sum the above inequality over states $s$:

$$\|\Delta\|_1 := \sum_s \Delta(s)$$

$$\leq \gamma \sum_s \sum_{s'} P(s \mid s', \pi_1(s'))\Delta(s') + \gamma \sum_s \sum_{s'} |P(s \mid s', \pi_1(s')) - P(s \mid s', \pi_2(s'))| \, d^{\pi_2}(s')$$

$$= \gamma \sum_{s'} \Delta(s') + \gamma \sum_{s'} d^{\pi_2}(s') \sum_s |P(s \mid s', \pi_1(s')) - P(s \mid s', \pi_2(s'))|$$

$$= \gamma \|\Delta\|_1 + \gamma \sum_{s'} d^{\pi_2}(s') \sum_s |P(s \mid s', \pi_1(s')) - P(s \mid s', \pi_2(s'))|. \tag{35}$$

Using the $K_P$-Lipschitz property for transition dynamics $P$:

$$\sum_s |P(s \mid s', \pi_1(s')) - P(s \mid s', \pi_2(s'))| \leq K_P \|\pi_1(s') - \pi_2(s')\|, \tag{36}$$

Substitute Eq. (36) into the TV bound in Eq. (35):

$$\|\Delta\|_1 \leq \gamma \|\Delta\|_1 + \gamma K_P \sum_{s'} d^{\pi_2}(s') \|\pi_1(s') - \pi_2(s')\| \tag{37}$$

$$\leq \gamma \|\Delta\|_1 + \gamma K_P \max_s \|\pi_1(s) - \pi_2(s)\|. \tag{38}$$

Rearranging terms to isolate $\|\Delta\|_1$, we obtain the following expression for the total variation distance between the state occupancy measures:

$$D_{\mathrm{TV}}(d^{\pi_1}, d^{\pi_2}) := \frac{1}{2}\|\Delta\|_1 \leq \frac{\gamma K_P}{2(1-\gamma)} \max_s \|\pi_1(s) - \pi_2(s)\|. \tag{39}$$

Finally, based on the definitions of the neighborhood and sample constraints, we conclude that for any deterministic policy $\pi_1$ that satisfies the neighborhood constraint (Definition 4), there exists a deterministic policy $\pi_2$ satisfying the sample constraint (Definition 3) such that for any $s \in \mathcal{D}$, $\|\pi_1(s) - \pi_2(s)\| \leq \epsilon$. For $s \notin \mathcal{D}$, We define $\pi_2(s) := \pi_1(s)$. Thus, $\max_s \|\pi_1(s) - \pi_2(s)\| \leq \epsilon$. Therefore, the TV distance between the state occupancies of such policies $\pi_1$ and $\pi_2$ satisfies:

$$D_{\mathrm{TV}}(d^{\pi_1}, d^{\pi_2}) \leq \frac{\gamma K_P \epsilon}{2(1-\gamma)}. \tag{40}$$

$\square$

Theorem 1 in the main text uses the standardness assumption, which is common in geometric measure theory [13, 8].

**Assumption 3** (Standardness, Assumption 1). *Let $S \subseteq \mathbb{R}^d$ be the support of a probability distribution $\nu$, and $B(x, r)$ be the closed ball of radius $r$ centered at $x \in \mathbb{R}^d$. There exist constants $r_0 > 0$ and $C_0 > 0$ such that:*

$$\forall x \in S, \ \forall r \leq r_0, \ \nu(B(x, r)) \geq C_0 \cdot r^d. \tag{41}$$

It ensures that the measure $\nu$ does not exhibit "holes" at small scales, preventing $\nu$ from being too sparse near any $x \in S$. It imposes a lower bound on the measure of small balls but does not explicitly require the existence of a density.

**Theorem 2** (Support approximation via neighborhoods, Theorem 1). *Let $S \subseteq \mathbb{R}^d$ be the compact support of a distribution $\nu$, and let $X_1, \ldots, X_n$ be independent and identically distributed samples from $\nu$. Define $U_{n,\epsilon} = \bigcup_{i=1}^n B(X_i, \epsilon)$ as the union of closed balls of radius $\epsilon$ centered at the samples. Let $\mathcal{N}(S, \epsilon/2)$ denote the covering number of $S$, i.e., the minimal number of $\epsilon/2$-balls required to cover $S$. Under the standardness Assumption 1 with constants $r_0, C_0 > 0$, for any $\delta \in (0, 1)$ and $\epsilon \leq 2r_0$, if*

$$n \geq \frac{1}{C_0(\epsilon/2)^d}\left(\log \mathcal{N}(S, \epsilon/2) + \log(1/\delta)\right), \tag{42}$$

*then with probability at least $1 - \delta$, the Hausdorff distance between $S$ and $U_{n,\epsilon}$ satisfies*

$$d_H(S, U_{n,\epsilon}) := \max\left(\sup_{x \in S} \inf_{u \in U_{n,\epsilon}} d(x, u), \ \sup_{u \in U_{n,\epsilon}} \inf_{x \in S} d(x, u)\right) \leq \epsilon. \tag{43}$$

*Proof.* By the compactness of $S$, for any $\epsilon > 0$, there exists a finite set of points $\{y_1, \ldots, y_N\} \subset S$ such that:

$$S \subseteq \bigcup_{j=1}^{N} B(y_j, \epsilon/2), \tag{44}$$

where $B(y_j, \epsilon/2)$ denotes the closed ball of radius $\epsilon/2$ centered at $y_j$, and $N = \mathcal{N}(S, \epsilon/2)$ denotes the covering number of $S$ for a given radius $\epsilon/2$, i.e., the minimal number of $\epsilon/2$-balls required to cover $S$. The existence of such a finite cover follows directly from the compactness of $S$.

For each $y_j$, the probability that none of the samples $X_1, \ldots, X_n$ lies in $B(y_j, \epsilon/2)$ is:

$$\mathbb{P}\left(B(y_j, \epsilon/2) \cap \{X_1, \ldots, X_n\} = \emptyset\right) = (1 - \nu(B(y_j, \epsilon/2)))^n. \tag{45}$$

Under the standardness assumption, there exist constants $r_0 > 0$ and $C_0 > 0$ such that

$$\forall x \in S, \ \forall r \le r_0, \ \nu(B(x,r)) \ge C_0 \cdot r^d. \tag{46}$$

For $\epsilon \le 2r_0$, we have $\epsilon/2 \le r_0$, and thus

$$\nu(B(y_j, \epsilon/2)) \ge C_0 \cdot (\epsilon/2)^d. \tag{47}$$

Using the inequality $1 - x \le e^{-x}$, it follows that

$$(1 - \nu(B(y_j, \epsilon/2)))^n \le e^{-n\nu(B(y_j,\epsilon/2))} \le e^{-nC_0(\epsilon/2)^d}. \tag{48}$$

Applying the union bound over all $N$ covering balls:

$$\mathbb{P}\left(\exists j \text{ s.t. } B(y_j, \epsilon/2) \cap \{X_1, \ldots, X_n\} = \emptyset\right) \le N \left(1 - \nu(B(y_j, \epsilon/2))\right)^n \le Ne^{-nC_0(\epsilon/2)^d}. \tag{49}$$

To ensure this probability is at most $\delta$, we require

$$Ne^{-nC_0(\epsilon/2)^d} := \mathcal{N}(S, \epsilon/2) \cdot e^{-nC_0(\epsilon/2)^d} \le \delta. \tag{50}$$

Solving for $n$, we obtain

$$n \ge \frac{\log \mathcal{N}(S, \epsilon/2) + \log(1/\delta)}{C_0(\epsilon/2)^d}, \tag{51}$$

which guarantees that, with probability at least $1 - \delta$, every $B(y_j, \epsilon/2)$ contains at least one sample $X_i$.

The Hausdorff distance is a metric used to quantify the discrepancy between two sets and is widely applied in shape comparison and analysis across various domains [29, 74]. It captures the maximum mismatch between the sets by measuring how far one must travel from a point in one set to reach the nearest point in the other. Mathematically, for two non-empty sets $A$ and $B$ in a metric space (e.g., Euclidean space) with distance function $d$, the Hausdorff distance $d_H(A, B)$ is defined as:

$$d_H(A,B) := \max \left( \sup_{a \in A} \inf_{b \in B} d(a,b), \sup_{b \in B} \inf_{a \in A} d(a,b) \right) \tag{52}$$

That is, for each point in set $A$, we find the closest point in set $B$, and take the maximum of these minimum distances. Similarly, for each point in set $B$, we find the closest point in set $A$, and take the maximum of these minimum distances. The Hausdorff distance is the larger of these two values.

This distance is sensitive to outliers, as even a single distant point can significantly increase the value of the distance. This property renders it particularly appropriate for evaluating approximation effects in our setting, where we aim to optimize the Q function over an approximated constraint set. Outliers in such a set can exert a substantial influence on the Q function's value.

The Hausdorff distance between $S$ and $U_{n,\epsilon} := \bigcup_{i=1}^{n} B(X_i, \epsilon)$ is:

$$d_H(S, U_{n,\epsilon}) := \max \left( \sup_{x \in S} \inf_{u \in U_{n,\epsilon}} d(x,u), \sup_{u \in U_{n,\epsilon}} \inf_{x \in S} d(x,u) \right) \tag{53}$$

where $d$ is the Euclidean distance.

We now analyze each term separately.

(1) Bounding the term $\sup_{x \in S} \inf_{u \in U_{n,\epsilon}} d(x, u)$:

By Eq. (44), for any $x \in S$, there exists $y_j$ such that $x \in B(y_j, \epsilon/2)$. By the coverage guarantee under Eq. (51) (with probability at least $1 - \delta$, every $B(y_j, \epsilon/2)$ contains at least one sample $X_i$), it follows that $B(y_j, \epsilon/2)$ contains some $X_i$. Thus,

$$d(x, X_i) \leq d(x, y_j) + d(y_j, X_i) \leq \epsilon/2 + \epsilon/2 = \epsilon. \tag{54}$$

This implies $x \in B(X_i, \epsilon) \subseteq U_{n,\epsilon}$. Since $x \in S$ is arbitrary, it holds that $S \subseteq U_{n,\epsilon}$, yielding $\sup_{x \in S} \inf_{u \in U_{n,\epsilon}} d(x, u) = 0$.

(2) Bounding the term $\sup_{u \in U_{n,\epsilon}} \inf_{x \in S} d(x, u)$:

By definition, any $u \in U_{n,\epsilon}$ belongs to $B(X_i, \epsilon)$ for some $X_i \in S$. Therefore, for any $u \in U_{n,\epsilon}$, the following holds:

$$\inf_{x \in S} d(x, u) \leq d(X_i, u) \leq \epsilon. \tag{55}$$

Thus, $\sup_{u \in U_{n,\epsilon}} \inf_{x \in S} d(x, u) \leq \epsilon$.

Combining both terms, we conclude

$$d_H(S, U_{n,\epsilon}) \leq \max(0, \epsilon) \leq \epsilon. \tag{56}$$

Therefore, when the sample size $n$ satisfies Eq. (51), it follows that, with probability at least $1 - \delta$, the Hausdorff distance between $S$ and $U_{n,\epsilon}$ satisfies $d_H(S, U_{n,\epsilon}) \leq \epsilon$. $\qquad\square$

## C  Experimental Details

### C.1  Experimental Details on D4RL Benchmarks

We adopt evaluation criteria consistent with those employed in prior studies. For the Gym locomotion tasks, performance is assessed by averaging returns over 10 evaluation trajectories and 5 random seeds. In the AntMaze suite, returns are averaged over 100 evaluation trajectories across the same number of seeds. Following the D4RL benchmark guidelines [20], we subtract 1 from the rewards in the AntMaze datasets. Additionally, in line with established practices [21, 40, 82, 88], we normalize the states in all Gym locomotion datasets. Our implementation builds upon TD3 [22], optimizing a deterministic policy. Consequently, we employ mean squared error in the weighted behavior cloning objective in Eq. (17), instead of a log-likelihood term. This approach is commonly used in RL algorithms, where a maximum likelihood problem is transformed into a regression problem when dealing with Gaussians with a fixed variance [21]. The architectures of the auxiliary policy and the final policy are identical, with the former's output range (i.e., max action) set to twice that of the latter to account for the most extreme cases. Performance is reported using normalized scores provided by the D4RL benchmark [20], which quantify the quality of the learned policy relative to both random and expert baselines:

$$\text{D4RL score} = 100 \times \frac{\text{learned policy return} - \text{random policy return}}{\text{expert policy return} - \text{random policy return}} \tag{57}$$

ANQ introduces two key hyperparameters: the Lagrange multiplier $\lambda$, which controls the overall neighborhood radius, and the inverse temperature $\alpha$, which controls the adaptiveness of neighborhood radius to advantage values. For the inverse temperature, we fix $\alpha = 1$ across all tasks, which yields consistently strong performance. For the Lagrange multiplier, we tune $\lambda$ within a small set of values $\{0.1, 5.0\}$ for all tasks. Given that part of our algorithm incorporates techniques from IQL [40], such as expectile regression and weighted behavior cloning, we adopt the hyperparameters suggested in their work: $\tau = 0.7$ and $\beta = 3$ for Gym locomotion tasks, and $\tau = 0.9$ and $\beta = 10$ for AntMaze tasks. To ensure that the constrained neighborhood is neither too large nor too small, we clip the exponentiated advantage weight $\exp(\alpha(Q_{\theta'}(s, a) - V_\psi(s)))$ in Eq. (13) to $[0.01, 30]$ in the Gym locomotion domains and $[0.01, 10]$ in the Antmaze domains. Following prior practices, we also clip the exponentiated weight $\exp(\beta(Q_{\theta'}(s, a + \mu_\omega(s, a)) - V_\psi(s)))$ in Eq. (17) to $[0, 3]$ in the Gym locomotion domains and $[0, 100]$ in the Antmaze domains to avoid instability. A comprehensive list of hyperparameter settings for ANQ is provided in Table 3.

**Insights into hyperparameter selection.** For selecting the Lagrange multiplier $\lambda$, a general principle is as follows: for narrow-distribution datasets (typically expert or demonstration data), a larger $\lambda$ can be used to induce smaller neighborhoods, thereby helping to more effectively control extrapolation error. For more dispersed datasets (often containing noisy or mixed-quality data), the impact of extrapolation error is relatively weaker; hence, a smaller $\lambda$ (i.e., larger neighborhoods) can be adopted to promote broader optimization over the action space and mitigate the impacts of suboptimal actions. Additionally, using a larger inverse temperature $\alpha$ in such cases can further enhance this effect by downweighting low-advantage actions more aggressively.

Table 3: Hyperparameters of ANQ.

| | Hyperparameter | Value |
|---|---|---|
| | Optimizer | Adam [37] |
| | Critic learning rate | $3 \times 10^{-4}$ |
| | Actor learning rate | $3 \times 10^{-4}$ with cosine schedule |
| | Discount factor | 0.99 for Gym, 0.995 for Antmaze |
| | Target update rate | 0.005 |
| ANQ | Policy update frequency | 2 |
| | Number of Critics | 4 |
| | Batch size | 256 |
| | Number of iterations | $10^6$ |
| | Lagrange multiplier $\lambda$ | $\{0.1, 5.0\}$ |
| | Inverse temperature $\alpha$ | 1 |
| IQL Specific | Expectile $\tau$ | 0.7 for Gym, 0.9 for Antmaze |
| | Inverse temperature $\beta$ | 3.0 for Gym, 10.0 for Antmaze |
| Architecture | Actor | input-256-256-output |
| | Critic | input-256-256-1 |

## C.2 Experimental Details of Noisy Data Experiments

In the noisy data experiments, we construct noisy datasets by mixing the random and expert datasets in D4RL at various expert ratios, thereby simulating real-world scenarios such as suboptimal data collection in autonomous systems or imperfect demonstrations in robotics. The total size of the combined dataset is fixed at $1 \times 10^6$. In certain environments, the sizes of the random or expert datasets in D4RL are slightly smaller than $1 \times 10^6$, so we directly use the corresponding D4RL dataset when the expert ratio is 0 or 1.

We evaluate the performance of several representative algorithms with different constraint types, including CQL (density constraint), IQL (sample constraint), SPOT (support constraint), and ANQ (neighborhood constraint). The hyperparameters of ANQ follow those used in the default benchmark datasets, as detailed in Table 3, and we fix $\lambda = 5$ for all the mixed datasets. In addition, we clip the exponentiated advantage weight in Eq. (13) to $[0.1, 100]$ for Hopper and Walker2d, and to $[0.1, 30]$ for Halfcheetah.

**Hyperparameters of baseline methods.** The hyperparameter configurations for the baseline methods are as follows. For CQL, we tune its regularization coefficient within the set $\{5, 10, 20, 30\}$, as it performs relatively well in this range, and report the best results obtained for each dataset. For IQL, we adopt the hyperparameters suggested in their work: expectile $\tau = 0.7$ and inverse temperature $\beta = 3$ on Gym locomotion tasks. For SPOT, we follow the implementation details provided in their paper, tuning its regularization coefficient within $\{0.05, 0.1, 0.2, 0.5, 1.0, 2.0\}$ on Gym locomotion tasks, and report the best results obtained for each dataset.

## C.3 Experimental Details of Limited Data Experiments

In the limited data experiments, we generate reduced datasets by randomly discarding some portion of transitions from the default AntMaze datasets. This setup simulates practical scenarios where data is sparse or partially missing, such as in healthcare applications.

We evaluate the performance of IQL (sample constraint), SPOT (support constraint), and ANQ (neighborhood constraint), excluding CQL (density constraint) due to its consistently inferior performance on Antmaze tasks, as reported in Table 2. The hyperparameters for ANQ follow those used in the default benchmark datasets, as detailed in Table 3, except that the Lagrange multiplier $\lambda$ is set to 2 to enable a sufficiently large neighborhood.

**Hyperparameters of baseline methods.** The hyperparameter configurations for the baseline methods are as follows. For CQL, we tune its regularization coefficient within the set $\{5, 10, 20, 30\}$, as it performs relatively well in this range, and report the best results obtained for each dataset. For IQL, we adopt the hyperparameters suggested in their work: expectile $\tau = 0.9$ and inverse temperature $\beta = 10$ on Antmaze tasks. For SPOT, we follow the implementation details provided in their paper, tuning its regularization coefficient within $\{0.025, 0.05, 0.1, 0.25, 0.5, 1.0\}$ on Antmaze tasks, and report the best results obtained for each dataset.

# D    Additional Experimental Results

## D.1    Computational Cost

We assess the runtime of offline RL algorithms on the halfcheetah-medium-replay-v2 dataset using a GeForce RTX 3090. The training time for ANQ and the baseline methods are presented in Figure 5. ANQ completes the task in approximately two hours, achieving competitive efficiency with other fast offline RL algorithms such as AWAC, IQL, and TD3BC,

Several factors contribute to this efficiency: (i) ANQ is model-free and ensemble-free, making it more efficient than model-based (e.g., MOPO [90]) and ensemble-based methods (e.g., EDAC [2]); (ii) ANQ uses simple three-layer MLPs for policy and value networks (as in TD3 [22]), which are more lightweight than complex architectures used in methods like Decision Transformer [10] or Diffusion Q-learning [80]; (iii) ANQ optimizes deterministic policies with a policy update frequency of 2 (as in TD3 [22]), offering slightly better efficiency than algorithms with stochastic policies; (iv) ANQ solves the bi-level objective via alternating updates, requiring only one extra lightweight optimization step for the auxiliary policy compared to IQL [40]. As a result, ANQ achieves competitive training efficiency while maintaining strong performance.

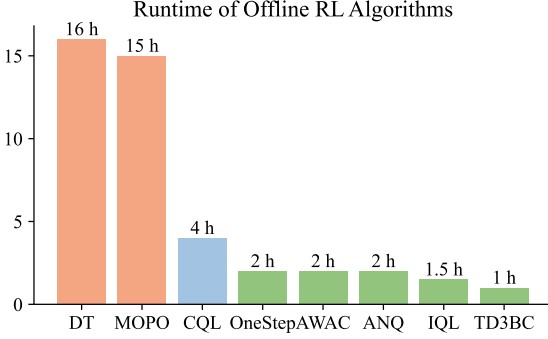

Figure 5: Runtime of algorithms on halfcheetah-medium-replay-v2 on a GeForce RTX 3090.

## D.2    Additional Benchmark Comparisons

This section extends our evaluation by comparing ANQ with additional recent SOTA algorithms of other constraint types on the D4RL benchmark [20]. For support constraints, we compare against STR [53], CPI [50], CPED [93], and SVR [54]. For sample constraints, we compare against IAC [92], SQL [88], and EQL [88]. As shown in Table 4, ANQ achieves the best overall score in the Antmaze domain and demonstrates highly competitive performance in the Gym locomotion domain. From a methodological design perspective, the proposed neighborhood constraint may be particularly beneficial in high-dimensional action spaces such as those in Antmaze. In such settings, support constraint methods often struggle to accurately model the behavior policy, while sample constraint

methods risk being overly conservative due to limited coverage of near-optimal actions. ANQ avoids both issues by enabling controlled generalization without explicit behavior policy modeling.

Table 4: Averaged normalized scores on Gym locomotion and Antmaze tasks over five random seeds. m = medium, m-r = medium-replay, m-e = medium-expert, e = expert, r = random; u = umaze, u-d = umaze-diverse, m-p = medium-play, m-d = medium-diverse, l-p= large-play, l-d = large-diverse.

| Dataset-v2 | IAC | SQL | EQL | STR | CPI | CPED | SVR | ANQ (Ours) |
|---|---|---|---|---|---|---|---|---|
| halfcheetah-m | 51.6±0.3 | 48.3±0.2 | 47.2±0.3 | 51.8±0.3 | **64.4±1.3** | 61.8±1.6 | 60.5±1.2 | 61.8±1.4 |
| hopper-m | 74.6±11.5 | 75.5±3.4 | 70.6±2.6 | **101.3±0.4** | 98.5±3.0 | **100.1±2.8** | 103.5±0.4 | **100.9±0.6** |
| walker2d-m | 85.2±0.4 | 84.2±4.6 | 83.2±4.4 | 85.9±1.1 | 85.8±0.8 | 90.2±1.7 | **92.4±1.2** | 82.9±1.5 |
| halfcheetah-m-r | 47.2±0.3 | 44.8±0.7 | 44.5±0.5 | 47.5±0.2 | 54.6±1.3 | **55.8±2.9** | 52.5±3.0 | **55.5±1.4** |
| hopper-m-r | **103.2±1.0** | 101.7±3.3 | 98.1±3.6 | 100.0±1.2 | 101.7±1.6 | 98.1±2.1 | 103.7±1.3 | 101.5±2.7 |
| walker2d-m-r | **93.2±1.8** | 77.2±3.8 | 81.6±4.2 | 85.7±2.2 | 91.8±2.9 | 91.9±0.9 | 95.6±2.5 | 92.7±3.8 |
| halfcheetah-m-e | 92.9±0.7 | **94.0±0.4** | **94.6±0.5** | 94.9±1.6 | 94.7±1.1 | 85.4±10.9 | 94.2±2.2 | **94.2±0.8** |
| hopper-m-e | 109.3±4.0 | **111.8±2.2** | **111.5±2.1** | **111.9±0.6** | 106.4±4.3 | 95.3±13.5 | 111.2±0.9 | 107.0±4.9 |
| walker2d-m-e | 110.1±0.1 | 110.0±0.8 | 110.2±0.8 | 110.2±0.1 | 110.9±0.4 | **113.0±1.4** | 109.3±0.2 | **111.7±0.2** |
| halfcheetah-e | 94.5±0.5 | - | - | 95.2±0.3 | **96.5±0.2** | - | **96.1±0.7** | 95.9±0.4 |
| hopper-e | 110.6±1.9 | - | - | **111.2±0.3** | **112.2±0.5** | - | 111.1±0.4 | 111.4±2.5 |
| walker2d-e | **114.8±1.2** | - | - | 110.1±0.1 | 110.6±0.1 | - | 110.0±0.2 | 111.8±0.1 |
| halfcheetah-r | 20.9±1.2 | - | - | 20.6±1.1 | **29.7±1.1** | - | 27.2±1.2 | 24.9±1.0 |
| hopper-r | **31.3±0.3** | - | - | **31.3±0.3** | 29.5±3.7 | - | **31.0±0.3** | **31.1±0.2** |
| walker2d-r | 3.0±1.3 | - | - | 4.7±3.8 | 5.9±1.7 | - | 2.2±1.5 | **11.2±9.5** |
| locomotion total | 1142.4 | - | - | 1162.2 | 1193.2 | - | **1200.5** | 1194.5 |
| antmaze-u | 77.6±3.8 | 92.2±1.4 | 93.2±2.2 | 93.6±4.0 | **98.8±1.1** | 96.8±2.6 | - | **96.0±1.6** |
| antmaze-u-d | 71.2±8.6 | 74.0±2.3 | 70.4±2.7 | 77.4±7.2 | **88.6±5.7** | 55.6±2.2 | - | 80.2±1.8 |
| antmaze-m-p | 72.0±7.6 | 80.2±3.7 | 77.5±4.3 | 82.6±5.4 | 82.4±5.8 | **85.1±3.4** | - | 76.2±3.3 |
| antmaze-m-d | 74.2±4.1 | 75.1±4.2 | 74.0±3.7 | **87.0±4.2** | 80.4±8.9 | 72.1±2.9 | - | 77.2±6.1 |
| antmaze-l-p | **57.0±7.4** | 50.2±4.8 | 45.6±4.2 | 42.8±8.7 | 20.6±16.3 | 34.9±5.3 | - | **56.2±4.9** |
| antmaze-l-d | 47.2±9.4 | **52.3±5.2** | 49.5±4.7 | 46.8±7.6 | 45.2±6.9 | 32.3±7.4 | - | **55.8±4.0** |
| antmaze total | 399.2 | 424.0 | 410.2 | 430.2 | 416.0 | 376.8 | - | **441.6** |

## D.3 Effect of the Auxiliary Policy

This section conducts an ablation study that replaces the auxiliary policy $\mu_\omega$ in ANQ with a random Gaussian noise.

**Discussion on exploiting vs. smoothing.** Defining the auxiliary policy $\mu$ as Gaussian noise is reminiscent of the target policy noise trick introduced in TD3 [22]. Specifically, in TD3, Gaussian noise is added to the policy actions in the Bellman target. Because the trained policy is prone to overfitting to local optima in the Q-function landscape, such noise injection can smooth the value estimates and reduce accumulated errors in Q-function training. By contrast, in our method, $\mu_\omega$ is applied to fixed in-dataset actions. Therefore, there is no need for Q-function estimate smoothing as in TD3. On the contrary, our method exploits, rather than smooths, the local Q landscape surrounding dataset actions, using it as a guide for the inner optimization. In our algorithm, $\mu_\omega$ is optimized to seek better actions within the neighborhood of dataset actions. Replacing $\mu_\omega$ with Gaussian noise thus turns this directed search into undirected random perturbations.

**Empirical results.** We compared different choices of the auxiliary policy $\mu$ on the Gym locomotion tasks, with results shown in Table 5. The tested variants include: (i) ANQ-Gaussian noise $\mu$: $\mu$ is replaced with Gaussian noise, i.e., $\mu \sim \text{clip}(\mathcal{N}(0, 0.04), -0.5, 0.5)$ as used in TD3 and TD3BC; (ii) IQL: correspond to ANQ with $\mu$ set to zero; (iii) ANQ-default: our default ANQ algorithm where $\mu_\omega$ is optimized. The results show that using an optimized $\mu$ (i.e., default ANQ) significantly outperforms the other two variants, especially on low-quality datasets, clearly demonstrating the benefit of our design. Additionally, ANQ with Gaussian noise $\mu$ performs slightly worse than IQL, likely because the Q values of dataset actions do not require smoothing, and the randomness introduced by Gaussian noise may cause the Bellman target to select actions inferior to the original dataset actions.

Table 5: Averaged normalized scores on Gym locomotion tasks over five random seeds.

| Dataset-v2 | IQL | ANQ-Gaussian noise $\mu$ | ANQ-stochastic $\mu\&\pi$ | ANQ-default |
|---|---|---|---|---|
| halfcheetah-m | 47.4±0.2 | 46.9±0.2 | 59.8±3.6 | 61.8±1.4 |
| hopper-m | 66.2±5.7 | 63.2±5.4 | 94.5±7.9 | 100.9±0.6 |
| walker2d-m | 78.3±8.7 | 82.9±2.2 | 84.3±0.9 | 82.9±1.5 |
| halfcheetah-m-r | 44.2±1.2 | 43.1±0.2 | 54.3±1.9 | 55.5±1.4 |
| hopper-m-r | 94.7±8.6 | 66.1±9.8 | 103.3±0.4 | 101.5±2.7 |
| walker2d-m-r | 73.8±7.1 | 81.2±4.5 | 93.6±2.9 | 92.7±3.8 |
| halfcheetah-m-e | 86.7±5.3 | 85.1±6.2 | 94.6±0.5 | 94.2±0.8 |
| hopper-m-e | 91.5±14.3 | 60.8±38.4 | 106.0±5.2 | 107.0±4.9 |
| walker2d-m-e | 109.6±1.0 | 111.9±0.5 | 112.1±0.2 | 111.7±0.2 |
| halfcheetah-r | 13.1±1.3 | 2.3±0.0 | 26.5±1.7 | 24.9±1.0 |
| hopper-r | 7.9±0.2 | 7.2±0.2 | 31.0±0.2 | 31.1±0.2 |
| walker2d-r | 5.4±1.2 | 6.0±0.2 | 8.3±7.6 | 11.2±9.5 |
| total | 718.8 | 656.7 | 868.2 | 875.4 |

### D.4 Stochastic Policies

In practice, our algorithm is compatible with stochastic policies. Both policies introduced in the method (the auxiliary policy $\mu$ and the final policy $\pi$) can be replaced by stochastic counterparts without modification to the overall framework.

For optimizing a stochastic auxiliary policy $\mu_\omega$ (typically Gaussian), Eq. (13) can be directly optimized using the reparameterization trick:

$$\max_{\mu_\omega} \mathbb{E}_{(s,a)\sim\mathcal{D}, \epsilon\sim\mathcal{N}(0,1)} \left[ Q_\theta(s, a + \mu_\omega(\epsilon; s, a)) - \lambda \exp(\alpha(Q_{\theta'}(s,a) - V_\psi(s))) \|\mu_\omega(\epsilon; s, a)\| \right].$$

For extracting a stochastic final policy $\pi_\phi$ (typically Gaussian), the regression problem in Eq. (17) can be reformulated as a maximum likelihood objective:

$$\min_{\pi_\phi} \mathbb{E}_{(s,a)\sim\mathcal{D}, \epsilon\sim\mathcal{N}(0,1)} \exp(\beta(Q_{\theta'}(s, a + \mu_\omega(\epsilon; s, a)) - V_\psi(s))) \log \pi_\phi(a + \mu_\omega(\epsilon; s, a)|s)$$

Empirically, we tested a stochastic version of the algorithm, denoted as ANQ-stochastic $\mu\&\pi$, where the both policies are Gaussian. The results, reported in Table 5, show that it performs comparably to the default ANQ across most Gym locomotion tasks, though slightly worse on a few.

### D.5 Learning Curves

Learning curves of ANQ on Gym-MuJoCo locomotion tasks and Antmaze tasks are presented in Figure 6 and Figure 7, respectively. The curves are averaged over 5 random seeds, with the shaded area representing the standard deviation across seeds.

## E Broader Impact

Offline reinforcement learning (RL) holds significant potential for expanding RL's real-world applications in fields like robotics, recommendation systems, healthcare, and education, particularly in scenarios where data collection is expensive or risky. Nevertheless, it is essential to be aware of the potential negative societal impacts that could arise from the deployment of offline RL systems. One concern is that the data used for training may contain inherent biases, which could then be reflected in the resulting policies, potentially exacerbating existing inequalities or reinforcing harmful stereotypes. Another issue is the potential impact of offline RL on employment, as it may lead to the automation of jobs that are currently performed by humans, such as in manufacturing or autonomous driving. To ensure the responsible use of offline RL, it is crucial to address these challenges, striking a balance between fostering innovation and mitigating adverse societal consequences.

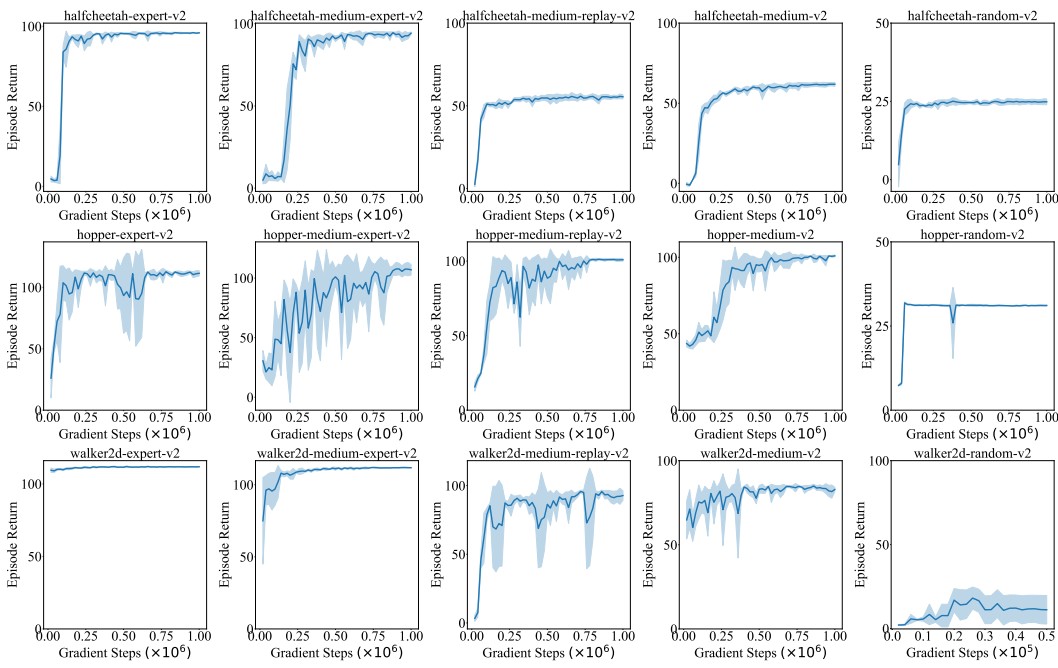

Figure 6: Learning curves of ANQ on Gym locomotion tasks during offline training. The curves are averaged over 5 random seeds, with the shaded area representing the standard deviation across seeds.

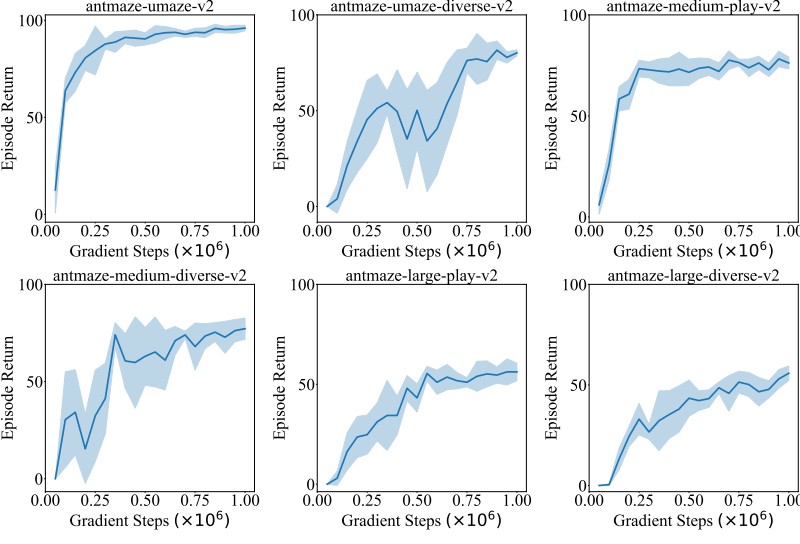

Figure 7: Learning curves of ANQ on Antmaze tasks during offline training. The curves are averaged over 5 random seeds, with the shaded area representing the standard deviation across seeds.

