# OpenReview forum: "Adaptive Neighborhood-Constrained Q Learning for Offline Reinforcement Learning"
_NeurIPS.cc/2025/Conference — NeurIPS 2025 spotlight_

### Official Review · Reviewer_EzpC · 2025-06-21

**Clarity:** 2
**Significance:** 2
**Originality:** 3
**Rating:** 4
**Confidence:** 4

**Summary:**

This paper proposed to tackle offline RL problems by enforcing neighborhood constraint rather than imposing density/support/sample constraints. Perturbations are added to action values $Q(s, \tilde{a})$ from an adaptive neighborhood such that $||a-\tilde{a}||\leq \epsilon\exp(-\alpha A(s,a))$. The authors formulated a bi-level optimization problem to first obtain $Q$ value from dataset actions then extract the policy from the perturbed Q value.

**Questions:**

Please refer to the weaknesses section for questions.

**Ethical Concerns:**

["NO or VERY MINOR ethics concerns only"]

**Final Justification:**

The paper presents a new constraint method to avoid OOD issue. I found the initial submission quite unclear and ambiguous, but the authors have addressed these concerns and promised will clean up the paper later. In summary, I believe these issues do not overweigh the paper's contributions. I therefore have raised my score.

**Limitations:**

yes

**Paper Formatting Concerns:**

I found no formatting issues.

**Quality:**

3

**Strengths And Weaknesses:**

**Strengths:**\
The motivation of developing a new type of constraint beyond the listed three classes is appreciated. I find the proposed method interesting and has potential links to several existing approaches.

**Weaknesses:**\
I found the paper quite hard to follow. Specifically, the first half of the technical sections (before Definition 5) and the second seem to be detached. I can understand that the motivation is to build a neighborhood set  $||a-\tilde{a}||\leq \epsilon$ such that (1) their value/distributional difference can be controlled; (2) generalization capability is guaranteed. To this end, the authors introduced several lemmas and propositions. However, these lemmas are either from existing works or already well-known. For example, lemma 1 is a very standard result and can be seen from e.g. [Achiam et al. 2017].  lemma 2 comes from [43]. I guess there are some issues with proposition 1 because by definition $D_{TV} \leq 1$ but the authors showed an upper bound depending on $|\mathcal{S}|$, which seems to be of little use even for small-sized problems.

The second half also needs more elaboration: Theorem 1 defines covering numbers in terms of $\mathcal{S}$. But shouldn't it be about actions since we care about neighbors and generalization capability of actions? If it is a typo and the theorem is indeed about actions, then for the assumption of "independent and identically distributed samples from $\nu$" to hold, the policy $\nu$ has to be fixed?


Definition 5 does not seem to be very well coupled with previous developments : the authors did not sufficiently motivate the radius  $epsilon\exp(-\alpha A(s,a))$ nor explained how it connects to the covering number theorem. More importantly, this quantity relies on accurate estimates of advantage in the offline context. But if we can have such accurate estimates free of OOD issues, in principle we don't have to employ this perturbation scheme?

---

> ### Author Rebuttal · Authors · 2025-07-31
>
> We appreciate the time and effort you have dedicated to providing feedback on our paper and are grateful for the meaningful comments.
>
> **Q1: The bound in Proposition 1 depends on $|\mathcal{S}|$.**
>
> Thank you for the valuable comment. We first clarify the original bound and then provide a tighter version that eliminates the $|\mathcal{S}|$-dependency.
>
> (1) Original bound: The bound in Proposition 1 is mathematically valid, but potentially loose. It scales linearly with the neighborhood radius $\epsilon$, which makes it meaningful when $\epsilon$ is small. However, the dependence on $|\mathcal{S}|$ can indeed make the bound less practical in large-scale settings. This linear dependence on $|\mathcal{S}|$ originates from the pointwise Lipschitz assumption on the transition dynamics, i.e.,
>
> $|P(s'|s,a_1) - P(s'|s,a_2)| \leq K_P \\|a_1 - a_2\\|, \quad \forall s,s' \in \mathcal{S},\forall a_1,a_2 \in \mathcal{A}$
>
> In the proof (Eq. (35)), this pointwise difference is summed over all possible next states, thus introducing the factor $|\mathcal{S}|$.
>
> (2) Tighter bound without $|\mathcal{S}|$-dependency:
>
> The $|\mathcal{S}|$-term can be effectively eliminated by replacing the pointwise Lipschitz assumption with an $L_1$-Lipschitz assumption on the transition dynamics:
>
> $\\|P(\cdot|s, a_1)-P(\cdot|s, a_2)\\| _ 1 \leq K_{L1} \\|a_1 - a_2\\|, \quad \forall s \in \mathcal{S},\forall a_1,a_2 \in \mathcal{A}$
>
> This assumption is consistent with prior theoretical RL studies [5], where bounding distributional shift under action perturbations is critical for convergence guarantees. A small $K_{L1}$ is also empirically supported by observations in physical systems, where small action changes typically lead to limited next-state variation [6].
>
> In the proof, we replace Eq. (35) with:
>
> $\sum_s \left| P(s \mid s', \pi_1(s')) - P(s \mid s', \pi_2(s')) \right| \leq K_{L1} \\|\pi_1(s') - \pi_2(s')\\|.$
>
> The rest of the derivation remains unchanged, leading to a revised bound:
>
> $\mathrm{D_{TV}}\left(d^{\pi_1}(\cdot), d^{\pi_2}(\cdot)\right) \leq \frac{\gamma K_{L1}}{2(1 - \gamma)} \epsilon$.
>
> This new bound is more broadly applicable for large-scale problems, as it is independent of $|\mathcal{S}|$ and typically yields smaller constants in real-world applications. We thank the reviewer for this observation and will include the new bound in the revision to further strengthen Proposition 1.
>
> **Q2: About the meaning of $S$ in Theorem 1 and fixed $\nu$.**
>
> We apologize for the confusion caused by the insufficient explanation of notations. Since both the support and neighborhood constraint set are defined over actions conditioned on a given state, Theorem 1 analyzes their approximation relationship at a fixed state, focusing on their difference in the action space. We adopt following abstract notations to formalize the analysis.
>
> > Theorem 1. Let $S \subseteq \mathbb{R}^d$ be the compact support of a distribution $\nu$, and let $X_1,\dots,X_n$ be i.i.d. samples from $\nu$. Define $U_{n,\epsilon} = \bigcup_{i=1}^n B(X_i, \epsilon)$ as the union of closed balls of radius $\epsilon$ centered at the samples. Let $\mathcal{N}(S, \epsilon/2)$ denote the covering number of $S$…
>
> In the context of offline RL: $\nu$ represents the behavior policy distribution on a given state, which is generally assumed fixed in offline RL. $S$ (distinct from the state space $\mathcal{S}$) is defined as the support of behavior policy $\nu$ at that state. $X_1,\dots,X_n$ are independent and identically distributed (i.i.d.) samples from the behavior policy $\nu$, corresponding to dataset actions at that state, consistent with the standard offline RL assumption of i.i.d. data from a fixed behavior policy. Thus, the theorem indeed concerns actions and $\nu$ is fixed.
>
> **Q3: Lemma 1 is a very standard result and can be seen from e.g. [Achiam et al. 2017]. Lemma 2 comes from [43].**
>
> Thank you for pointing this out. We agree that Lemma 1 and Lemma 2 are based on established results. Lemma 1 is derived from the performance difference lemma [1] and is related to results in TRPO [2] and CPO [3]. Lemma 2 is a direct corollary of Lemma 4 in [4], specialized to the case of action extrapolation, as noted in the proof. These lemmas are included not as novel contributions, but to highlight the limitations of existing approaches and to motivate the design of our method. We present them as lemmas (rather than propositions or theorems) to distinguish them from our contributions. However, we acknowledge that their connections to prior work were not sufficiently explained in the main text, which could cause confusion. We will revise the manuscript to clarify these points and properly attribute the prior results. The main theoretical contributions of our work are captured in the proposition and theorem.
>
> **Q4: The first half of the technical sections (before Definition 5) and the second seem to be detached.**
>
> Thanks for the comment. The flow of the technical sections is as follows:
>
> (1) The first half (before Definition 5) analyzes the limitations of existing constraint types, proposes neighborhood constraint as a new type, and establishes its theoretical properties under uniform neighborhood radii.
>
> (2) The second half (after Definition 5) develops a tractable optimization algorithm that enables Q learning under the proposed neighborhood constraint. Notably, although the optimization algorithm is presented specifically for the adaptive neighborhood in Definition 5, i.e., using radius $\epsilon \exp(-\alpha A(s,a))$, the framework is general and can accommodate any adaptive radius schemes. Specifically, one can simply replace $\epsilon \exp(-\alpha A(s,a))$ in Definition 5 (and Eq. (11)) with $\epsilon f(s,a)$ to define arbitrary per-sample neighborhood radii, where $f: \mathcal S \times \mathcal A \to \mathbb R^+$ is any function adjusting the radius. Correspondingly, in Eq. (12), $\exp(\alpha A(s,a))$  becomes $1/f(s,a)$, and Eq. (13) becomes:
>
> $\max_{\mu_\omega} \mathbb E_{(s,a)\sim \mathcal{D}} [Q_\theta(s,a+\mu_\omega(s,a)) - \lambda \\|\mu_\omega(s,a)\\| / f(s,a)].$
>
> All other equations remain unchanged, yielding an algorithm that supports arbitrary neighborhood radius schemes.
>
> (3) Definition 5 serves as a transition point between the theoretical and practical parts. It provides a concrete instantiation of adaptive neighborhoods, where the radius is adjusted based on estimated advantage values, motivated by empirical insights.
>
> **Q5: About Definition 5 and advantage estimates.**
>
> > Definition 5 does not seem to be very well coupled with previous developments: the authors did not sufficiently motivate the radius $\epsilon \exp(-\alpha A(s,a))$ nor explained how it connects to the covering number theorem.
>
> (1) Definition 5 is introduced as a practical instantiation of adaptive neighborhoods, where the sample-wise radius is set as $\epsilon \exp(-\alpha A(s,a))$ based on estimated advantages. As explained in the paragraph preceding Definition 5, this design is motivated by empirical insights: expert-like actions are typically concentrated and require smaller neighborhood to reduce extrapolation, while suboptimal actions tend to be more dispersed and can tolerate and benefit from larger neighborhood to enable broader search. The exponential form of advantages $\epsilon \exp(-\alpha A(s,a))$ heuristically captures this intuition.
>
> (2) Definition 5 is intended as a practical complement based on the theoretical foundation established under the uniform setting. Theorem 1 establishes theoretical guarantees under uniform neighborhood radii, serving as a foundation for understanding geometric coverage. Building on this foundation, Definition 5 extends the formulation by introducing an adaptive radius scheme guided by empirical insights. It serves as an empirically grounded instantiation that fits naturally within our framework and supports the broader goal of enabling flexible, pointwise conservatism.
>
> > More importantly, this quantity ($\epsilon \exp(-\alpha A(s,a))$) relies on accurate estimates of advantage in the offline context. But if we can have such accurate estimates free of OOD issues, in principle we don't have to employ this perturbation scheme?
>
> Thanks for the comment. We will clarify from two perspectives:
>
> (1) In-distribution estimation: This quantity (adaptive radius) relies solely on advantage estimates at dataset points $(s,a) \in \mathcal D$, which are in-distribution (in fact, in-sample) and therefore more reliable than estimates at OOD points.
>
> (2) No need for precise advantage: The purpose of using advantage is to qualitatively distinguish between good and suboptimal actions, and exact advantage values are not required. The exponential form $\epsilon \exp(-\alpha A(s,a))$ is a soft heuristic to bias the neighborhood radius. Our motivation does not assume or require the radius to be exactly proportional to the exponential of the advantage.
>
> **Reference**
>
> [1] Kakade et al. Approximately optimal approximate reinforcement learning. ICML 2002.
>
> [2] Schulman et al. Trust region policy optimization. ICML 2015.
>
> [3] Achiam et al. Constrained policy optimization. ICML 2017.
>
> [4] Li et al. When Data Geometry Meets Deep Function: Generalizing Offline Reinforcement Learning. ICLR 2023.
>
> [5] Asadi et al. Lipschitz continuity in model-based reinforcement learning. ICML 2018.
>
> [6] Todorov et al. Mujoco: A physics engine for model-based control. IROS 2012.

---

### Official Review · Reviewer_5EYf · 2025-07-02

**Clarity:** 4
**Significance:** 3
**Originality:** 3
**Rating:** 5
**Confidence:** 4

**Summary:**

This paper attempts to address the extrapolation error issue resulting from OOO actions in offline RL by restricting action selection for the Bellman target within some neighborhood of actions in the dataset. The authors provide theoretical guarantees for this choice of constraints. Practically, the authors propose an efficient algorithm by decomposing the objective into a bilevel optimization problem.

**Questions:**

See question above on stochastic policies.

**Ethical Concerns:**

["NO or VERY MINOR ethics concerns only"]

**Limitations:**

Yes

**Quality:**

3

**Strengths And Weaknesses:**

Strength

- The paper is very well written and easy to read.
- I particularly like the summary of different constraint types used in modern offline RL algorithms, the authors did an excellent job in summarizing and categorizing recent works with different constraint types, as well as addressing their strengths and limitations.
The idea of using neighborhood constraints is straightforward and yet powerful, the authors did a good job at motivating this.
- I think the theoretical analysis is also one of the key strengths of the paper, with the authors showing through Theorem 1 that the union of sample neighborhoods can approximate the true support.
- The bilevel optimization algorithm proposed is sound. Using the advantage function for the adaptive constraints is probably the most intuitive choice here and appeared to show good results. I agree with the authors comments in the limitations section that it is worth exploring different instantiations of adaptive neighborhoods in future works.
- Experiments are very well done with an extensive set of environments and baselines, as well as very detailed comparisons in terms of performance and compute time with each. Empirical results are quite strong

Weaknesses

- The theoretical analysis as well as the practical algorithm assumes a deterministic policy, could the authors comment how/whether it is possible to extend to stochastic policies?
- The algorithm itself seems to be quite sensitive to the choice of $\lambda$ and $\alpha$, however the authors are very up front about this and have performed extensive ablation studies on these hyperparameters, as well as giving nice interpretations on the values for $\lambda$ and $\alpha$.

---

> ### Author Rebuttal · Authors · 2025-07-31
>
> We appreciate the time and effort you have dedicated to providing feedback on our paper and are grateful for the meaningful comments.
>
> **Q1: Could the authors comment how/whether it is possible to extend to stochastic policies?**
>
> Thank you for the meaningful question.
>
> (1) From a practical perspective, our algorithm is fully compatible with stochastic policies. Both policies introduced in the method (the auxiliary policy and the final policy) can be replaced by stochastic counterparts without modification to the core framework.
>
> - For optimizing a stochastic auxiliary policy $\mu_\omega$, Eq. (13) can be directly optimized using the reparameterization trick:
>
> $\max_{\mu_\omega} \mathbb{E}_ {(s,a)\sim \mathcal{D},\epsilon\sim\mathcal{N}(0,1)} \left[ Q_\theta(s,a+\mu_\omega(\epsilon;s,a)) - \lambda \exp(\alpha (Q_{\theta'}(s,a) - V_\psi(s))) \\|\mu_\omega(\epsilon;s,a)\\| \right].$
>
> - For extracting a stochastic final policy $\pi_\phi$, the regression problem in Eq.(16) can be reformulated as a maximum likelihood problem:
>
> $\min_{\pi_\phi} \mathbb{E}_ {(s,a)\sim \mathcal{D},\epsilon\sim\mathcal{N}(0,1)} \exp (\beta(Q_{\theta'}(s,a+\mu_\omega(\epsilon;s,a)) - V_\psi(s))) \log \pi_\phi(a+\mu_\omega(\epsilon;s,a)|s)$
>
> (2) Empirically, we tested a stochastic version of the algorithm, denoted as ANQ-Gaussian, where both policies are Gaussian. The results, reported in the table below, show that ANQ-Gaussian performs comparably to the original ANQ across most Gym locomotion tasks, though slightly worse on a few.
>
> Table 1: Averaged normalized scores on Gym locomotion tasks over 5 random seeds.
>
> | Dataset-v2 | ANQ-Gaussian | ANQ |
> |---|---|---|
> | halfcheetah-m | 59.8$\pm$3.6 | **61.8$\pm$1.4** |
> | hopper-m | 94.5$\pm$7.9 | **100.9$\pm$0.6** |
> | walker2d-m | **84.3$\pm$0.9** | 82.9$\pm$1.5 |
> | halfcheetah-m-r | 54.3$\pm$1.9 | **55.5$\pm$1.4** |
> | hopper-m-r | **103.3$\pm$0.4** | 101.5$\pm$2.7 |
> | walker2d-m-r | **93.6$\pm$2.9** | 92.7$\pm$3.8 |
> | halfcheetah-m-e | **94.6$\pm$0.5** | 94.2$\pm$0.8 |
> | hopper-m-e | 106.0$\pm$5.2 | **107.0$\pm$4.9** |
> | walker2d-m-e | **112.1$\pm$0.2** | 111.7$\pm$0.2 |
> | halfcheetah-r | **26.5$\pm$1.7** | 24.9$\pm$1.0 |
> | hopper-r | 31.0$\pm$0.2 | **31.1$\pm$0.2** |
> | walker2d-r | 8.3$\pm$7.6 | **11.2$\pm$9.5** |
> | total | 868.2 | **875.4** |
>
> (3) Theoretically, while Proposition 1 relies on a deterministic policy, Lemma 2 and Theorem 1 hold independently of policy determinism, as they concern properties of the neighborhood constraint itself. However, in practice, a stochastic policy may occasionally generate actions that violate the neighborhood constraint, making it only probabilistically satisfied. Investigating the theoretical properties of probabilistic neighborhood constraints presents an interesting direction for future work.
>
> **Q2: Sensitivity to $\lambda$ and $\alpha$.**
>
> We appreciate the comment and will discuss the hyperparameter sensitivity in three aspects:
>
> (1) Superior performance within moderate parameter ranges. As shown in the ablation study, ANQ generally exhibits stronger and more robust performance with intermediate values of Lagrange multiplier $\lambda$ and inverse temperature $\alpha$. Since $\lambda=0$ and $\lambda=\infty$ correspond to no constraint and sample constraint respectively, the results indicate that the proposed neighborhood constraint outperforms both across a range of hyperparameters. Similarly, as $\alpha=0$ corresponds to a fixed neighborhood radius, the results show that advantage-based adaptive neighborhoods generally outperform uniform neighborhoods.
>
> (2) Relatively limited hyperparameter tuning. The specific ANQ algorithm involves relatively little hyperparameter tuning. As detailed in Appendix C1, we fix $\alpha=1$ across all tasks, which yields consistently strong performance. For $\lambda$, we use only a small set of values: $\lambda = 5$ for Antmaze, and $\lambda \in \\{0.1, 5\\}$ for Gym locomotion, involving less tuning compared to baselines like CQL and SPOT.
>
> (3) Insights into hyperparameter selection. For selecting $\lambda$, a general principle is as follows: for narrow-distribution datasets (typically expert or demonstration data), a larger $\lambda$ can be used to induce smaller neighborhoods, thereby helping to more effectively control extrapolation error. For more dispersed datasets (often containing noisy or mixed-quality data), the impact of extrapolation error is relatively weaker; hence, a smaller $\lambda$ (i.e., larger neighborhoods) can be adopted to promote broader optimization over the action space and mitigate the impacts of suboptimal actions. Additionally, using a larger $\alpha$ in such cases can further enhance this effect by downweighting low-advantage actions more aggressively.

---

> > ### Comment · Reviewer_5EYf · 2025-08-08
> >
> > Thank you for taking the time with the detailed response and extra experiments, please be sure to add them to the next revision of the paper, after reading through the response and the reviews from other reviewers, I stand by my original evaluation, good job!

---

> > > ### Author Response · Authors · 2025-08-08
> > >
> > > Thank you very much for your support and constructive feedback. We will make sure to include the additional results in the revised manuscript.

---

> ### Comment · Area_Chair_Vvee · 2025-08-06
>
> Please remember to respond to authors' rebuttal as soon as possible.
>
> Thank you!
>
> -AC

---

### Official Review · Reviewer_DsUf · 2025-07-02

**Clarity:** 4
**Significance:** 4
**Originality:** 3
**Rating:** 6
**Confidence:** 5

**Summary:**

This paper addresses the challenge of extrapolation errors in offline reinforcement learning (RL), which occur when evaluating out-of-distribution (OOD) actions not present in the static training dataset. The authors propose a novel neighborhood constraint that restricts action selection in the Bellman target to the union of neighborhoods around dataset actions, offering a flexible middle ground between overly conservative density/sample constraints and the computationally demanding support constraint (which requires accurate behavior policy modeling). The proposed constraint bounds extrapolation errors and distribution shift while approximating the least restrictive support constraint without requiring behavior policy modeling. The constraint is made pointwise adaptive by adjusting the neighborhood radius for each data point based on its advantage (quality). Empirically, ANQ achieves state-of-the-art results on standard offline RL benchmarks (D4RL), outperforming existing methods like CQL, IQL, and SPOT on both Gym locomotion and challenging AntMaze tasks.

**Questions:**

* Could u please provide an additional ablation that changes the $\mu_{\omega}$ to a random variable sampled from gaussion?  As the performance improvement could also be derived from the Q-landscape smoothness despite theoretical supports.

**Ethical Concerns:**

["NO or VERY MINOR ethics concerns only"]

**Final Justification:**

I think the authors have fully addressed my concerns. I recommend to accept the paper.

**Limitations:**

yes

**Quality:**

4

**Strengths And Weaknesses:**

## Strength
* The paper introduces a neighborhood constraint that restricts Bellman targets to the union of neighborhoods around dataset actions. Theoretical analyses (Lemma 2, Proposition 1, Theorem 1) rigorously establish that the neighborhood constraint bounds extrapolation errors under mild continuity assumptions, approximates the support constraint without needing behavior policy modeling and
controls distribution shift via Lipschitz continuity of transition dynamics.
* The proposed bilevel optimization (inner maximization + outer expectile regression) efficiently enforces the constraint during Q-learning.
* ANQ achieves top-tier results on D4RL benchmarks, including Gym locomotion and challenging AntMaze tasks. Robustness is validated in noisy data and limited data scenarios, where ANQ outperforms density (CQL), sample (IQL), and support (SPOT) constraints.

## Weakness
* The performance of ANQ might be sensitive to the choice of hyperparameters. A more detailed analysis of hyperparameter sensitivity would be beneficial to understand the robustness of the method.
* Some new sota support/sample constraint baselines are suggested such as [1][2].


[1] Iteratively Refined Behavior Regularization for Offline Reinforcement Learning

[2] Supported trust region optimization for offline reinforcement learning

---

> ### Author Rebuttal · Authors · 2025-07-31
>
> We appreciate the time and effort you have dedicated to providing feedback on our paper and are grateful for the meaningful comments.
>
> **Q1: A more detailed analysis of hyperparameter sensitivity would be beneficial to understand the robustness of the method.**
>
> Thank you for the suggestion. We analyze the hyperparameter sensitivity from the following three aspects:
>
> (1) Superior performance within moderate parameter ranges. As shown in the ablation study in Section 4.4, ANQ generally exhibits stronger and more robust performance with intermediate values of Lagrange multiplier $\lambda$ and inverse temperature $\alpha$. Since $\lambda=0$ and $\lambda=\infty$ correspond to no constraint and sample constraint respectively, the results indicate that the proposed neighborhood constraint outperforms both across a range of hyperparameters. Similarly, as $\alpha=0$ corresponds to a fixed neighborhood radius, the results show that advantage-based adaptive neighborhoods generally outperform uniform neighborhoods.
>
> (2) Relatively limited hyperparameter tuning. The specific ANQ algorithm involves relatively little hyperparameter tuning. As detailed in Appendix C1, we fix $\alpha=1$ across all tasks, which yields consistently strong performance. For $\lambda$, we use only a small set of values: $\lambda = 5$ for Antmaze, and $\lambda \in \\{0.1, 5\\}$ for Gym locomotion, involving less tuning compared to baselines like CQL and SPOT.
>
> (3) Insights into hyperparameter selection. For selecting $\lambda$, a general principle is as follows: for narrow-distribution datasets (typically expert or demonstration data), a larger $\lambda$ can be used to induce smaller neighborhoods, thereby helping to more effectively control extrapolation error. For more dispersed datasets (often containing noisy or mixed-quality data), the impact of extrapolation error is relatively weaker; hence, a smaller $\lambda$ (i.e., larger neighborhoods) can be adopted to promote broader optimization over the action space and mitigate the impacts of suboptimal actions. Additionally, using a larger $\alpha$ in such cases can further enhance this effect by downweighting low-advantage actions more aggressively.
>
> **Q2: Some new sota support/sample constraint baselines are suggested such as [1][2].**
>
> Thanks for the suggestion. The table below reports the overall performance of STR, CPI, and ANQ. ANQ achieves the best results in the Antmaze domain and demonstrates competitive performance with CPI in the Gym locomotion domain.  A full comparison will be included in the revised manuscript.
>
> Table 2: Total scores on Gym locomotion and Antmaze tasks, averaged over 5 random seeds.
>
> | Dataset-v2 | STR | CPI | ANQ |
> |---|---|---|---|
> | locomotion total | 1162.2 | 1193.2 | **1194.5** |
> | antmaze total | 430.2 | 416.0 | **441.6** |
>
> **Q3: Ablation that changes $\mu_\omega$ to a random variable sampled from Gaussian.**
>
> Thanks for the insightful question.
>
> (1) Clarification on exploiting rather than smoothing. Defining the auxiliary policy $\mu$ as Gaussian noise is reminiscent of the target policy noise trick introduced in TD3. Specifically, in TD3, Gaussian noise is added to the policy actions in the Bellman target. Because the trained policy is prone to overfitting local optima in the Q-function landscape, such noise injection can smooth the value estimates and reduce accumulated errors in Q-function training. By contrast, in our method, $\mu_\omega$ is applied to fixed in-dataset actions. Therefore, there is no need for Q-function estimate smoothing as in TD3. On the contrary, our method exploits, rather than smooths, the local Q landscape surrounding dataset actions, using it as a guide for the inner optimization. In our algorithm, $\mu_\omega$ is optimized to seek better actions within the neighborhood of dataset actions. Replacing $\mu_\omega$ with Gaussian noise essentially turns this directed search into undirected random perturbations. Thus, our method is not intended to smooth the Q landscape, but rather to exploit its local structure for improvement.
>
> (2) Empirical results. We compared different choices of the auxiliary policy $\mu$ on the Gym locomotion tasks. Results are shown in the table below. The tested variants include:
>
> - ANQ w/o $\mu$: $\mu$ is set to zero, equivalent to IQL.
> - ANQ w/ Gaussian noise $\mu$: $\mu$ is replaced with Gaussian noise, i.e., $\mu \sim \text{clip}(\mathcal{N}(0, 0.04), -0.5, 0.5)$ as used in TD3 and TD3+BC.
> - ANQ w/ optimized $\mu$: our default ANQ algorithm where $\mu_\omega$ is optimized.
>
> Table 1: Averaged normalized scores on Gym locomotion tasks over 5 random seeds.
>
> | Dataset-v2 | ANQ w/o $\mu$ (IQL) | ANQ w/ Gaussian noise $\mu$ | ANQ w/ optimized $\mu$ |
> |---|---|---|---|
> | halfcheetah-m | 47.4$\pm$0.2 | 46.9$\pm$0.2 | **61.8$\pm$1.4** |
> | hopper-m | 66.2$\pm$5.7 | 63.2$\pm$5.4 | **100.9$\pm$0.6** |
> | walker2d-m | 78.3$\pm$8.7 | 82.9$\pm$2.2 | **82.9$\pm$1.5** |
> | halfcheetah-m-r | 44.2$\pm$1.2 | 43.1$\pm$0.2 | **55.5$\pm$1.4** |
> | hopper-m-r | 94.7$\pm$8.6 | 66.1$\pm$9.8 | **101.5$\pm$2.7** |
> | walker2d-m-r | 73.8$\pm$7.1 | 81.2$\pm$4.5 | **92.7$\pm$3.8** |
> | halfcheetah-m-e | 86.7$\pm$5.3 | 85.1$\pm$6.2 | **94.2$\pm$0.8** |
> | hopper-m-e | 91.5$\pm$14.3 | 60.8$\pm$38.4 | **107.0$\pm$4.9** |
> | walker2d-m-e | 109.6$\pm$1.0 | **111.9$\pm$0.5** | 111.7$\pm$0.2 |
> | halfcheetah-r | 13.1$\pm$1.3 | 2.3$\pm$0.0 | **24.9$\pm$1.0** |
> | hopper-r | 7.9$\pm$0.2 | 7.2$\pm$0.2 | **31.1$\pm$0.2** |
> | walker2d-r | 5.4$\pm$1.2 | 6.0$\pm$0.2 | **11.2$\pm$9.5** |
> | total | 718.8 | 656.7 | **875.4** |
>
> The results show that using an optimized $\mu$ significantly outperforms the other two variants, especially on low-quality datasets, clearly demonstrating the benefit of our design. Additionally, ANQ with Gaussian noise performs slightly worse than ANQ without $\mu$, likely because the Q values of dataset actions do not require smoothing, and the randomness introduced by Gaussian noise may cause the Bellman target to select actions inferior to the original dataset actions.

---

> > ### Comment · Reviewer_DsUf · 2025-08-05
> >
> > I think my concerns have been addressed. Please include the new results in the revised paper. Based on the comments of all reviewers and the responses from the authors, I have decided to revise my score.

---

> > > ### Author Response · Authors · 2025-08-05
> > >
> > > Thank you very much for your constructive feedback. We will ensure that the new results are incorporated into the revised manuscript. If there are any remaining concerns or questions, we would be more than happy to address them within the available time.

---

### Official Review · Reviewer_JKyo · 2025-07-05

**Clarity:** 3
**Significance:** 2
**Originality:** 2
**Rating:** 5
**Confidence:** 3

**Summary:**

The paper introduces Adaptive Neighborhood-Constrained Q Learning (ANQ), a method designed to mitigate extrapolation bias in offline reinforcement learning (RL) by applying an adaptive neighborhood constraint to restrict action selection within a set of actions that are near the ones seen in the dataset. This constraint is enforced using a bi-level optimization framework, where the inner optimization updates the Q-function and the outer optimization updates the policy parameters.

**Questions:**

1. The bi-level optimization framework introduces significant complexity by requiring four separate updates per mini-batch. This could lead to instability or slow convergence, especially in environments with high variance or limited data. Could the authors provide a more detailed analysis of how this affects training stability and convergence in practice? Are there any strategies for mitigating potential instability, particularly when training with limited data or in environments with high-dimensional state-action spaces?

2. Given the increased complexity of bi-level optimization, does the method show diminishing returns as the training progresses? How does the computational cost of this approach compare to other state-of-the-art offline RL methods, especially in terms of training time and memory usage? Could the method be optimized further to reduce the computational burden?

3. The Lagrange multiplier λ and the inverse temperature α play important roles in controlling the neighborhood radius and enforcing the constraint. Could the authors provide more insight into how these hyperparameters are tuned in practice? Does their choice significantly impact training stability or the final policy performance? Is there an automatic way to tune these parameters based on the environment or data?
Scalability to Complex and High-Dimensional Tasks:
While the method performs well in standard benchmark tasks, how does it scale to high-dimensional action spaces or environments with partially observable states? Does the method retain its robustness and performance in such scenarios, or are modifications needed to accommodate the complexity of real-world tasks?

**Ethical Concerns:**

["NO or VERY MINOR ethics concerns only"]

**Final Justification:**

The rebuttal discussion solves me concerns.

**Limitations:**

Yes

**Paper Formatting Concerns:**

Nan

**Quality:**

3

**Strengths And Weaknesses:**

**Strengths:**

1. The introduction of an adaptive neighborhood constraint is a novel and effective approach for balancing conservatism and exploration in offline RL.

2. The method addresses a significant problem in offline RL, providing a flexible solution to extrapolation bias without the need for complex behavior policy modeling.

3. The paper is generally well-written, with clear motivations, method descriptions, and results. The authors effectively explain the problem of extrapolation bias in offline RL and introduce the adaptive neighborhood constraint as a solution.

**Weaknesses:**

1. Bi-level optimization introduces significant complexity to the method. Since each mini-batch requires four separate updates to the parameters, the computational cost becomes expensive. This results in increased training time per iteration, which could be a significant drawback, especially in high-dimensional or real-world applications.

2. The instability introduced by the bi-level optimization process is not fully explored. Given that each mini-batch requires multiple updates to the parameters, this could lead to issues with convergence or training stability, particularly in environments with high variance or limited data.

3. A more detailed discussion of the potential instability and its mitigation would provide clearer guidance for users and researchers adopting this method.

---

> ### Author Rebuttal · Authors · 2025-07-31
>
> We appreciate the time and effort you have dedicated to providing feedback on our paper and are grateful for the meaningful comments.
>
> **Q1: About the cost and complexity of bi-level optimization.**
>
> (1) Runtime efficiency.
>
> We briefly describe the training time results in the Benchmark Results section, with detailed comparisons provided in Figure 5 of Appendix D.1. The proposed ANQ is among the fastest tier of offline RL algorithms, on par with efficient baselines such as AWAC, IQL, and TD3BC. Several factors contribute to this efficiency:
>
> - ANQ is model-free and ensemble-free, making it more efficient than model-based (e.g., MOPO [1]) and ensemble-based methods (e.g., EDAC [2]);
> - ANQ uses simple three-layer MLPs for policy and value networks (as in TD3), which are more lightweight than complex architectures used in methods like Decision Transformer [3] or Diffusion Q-learning [4];
> - ANQ optimizes deterministic policies with a policy update frequency of 2 (as in TD3), offering slightly better efficiency than algorithms with stochastic policies;
> - ANQ solves the bi-level objective via alternating updates, requiring only one extra lightweight optimization step for the auxiliary policy compared to IQL.
>
> As a result, ANQ achieves competitive training efficiency while maintaining strong performance.
>
> (2) Memory usage.
>
> All four networks used in ANQ are three-layer MLPs, resulting in low memory overhead as well. We test GPU memory usage on the halfcheetah-medium-replay-v2 dataset using a GeForce RTX 3090.  As shown in the table below, the memory consumption of ANQ is also comparable with other efficient offline RL algorithms.
>
> Table 2: GPU memory usage of offline RL algorithms.
>
> | Algorithm | CQL | TD3BC | EDAC | ANQ |
> | --- | --- | --- | --- | --- |
> | GPU Memory (GB) | 1.4 | 1.4 | 1.9 | 1.4 |
>
> **Q2: About potential instability from bi-level optimization.**
>
> Thanks for the thoughtful comment. As shown in the training curves (Appendix D.2, Figures 6 and 7), ANQ generally demonstrates stable learning and convergence across most tasks. We observe no notable signs of instability such as divergence or diminishing returns.
>
> > this could lead to issues with convergence or training stability, particularly in environments with high variance or limited data.
>
> (1) High-variance environments. Hopper and Walker2d are commonly used to evaluate robustness in high-variance environments due to their sensitive dynamics. Even small action changes can cause the agent to fall, leading to highly variable returns. Our results show that ANQ maintains relatively stable training progress and strong convergence performance on both tasks, suggesting that the bi-level optimization framework does not compromise stability in these high-variance environments.
>
> (2) Limited data settings. It is worth noting that Section 4.3 specifically evaluates performance under limited data. As shown in Figure 2, ANQ consistently outperforms baselines with more stable final performance across varying data discard ratios, supporting its robustness in limited data scenarios.
>
> > A more detailed discussion of the potential instability and its mitigation would provide clearer guidance for users and researchers adopting this method.
>
> While our results demonstrate ANQ's overall stability across tasks, we agree that guidance on mitigating potential instability can be valuable. Below are two practical suggestions based on our analysis.
>
> (1) Increase $\lambda$: Ablation results show that too small $\lambda$ (i.e., too large neighborhoods) can lead to increased return variance. Therefore, increasing $\lambda$, which tightens the neighborhood constraint, may help enhance stability if necessary, though potentially at the cost of reduced policy flexibility.
>
> (2) Decrease $\alpha$: Although ANQ remains stable across a wide range of $\alpha$ in our experiments, we hypothesize that setting a smaller $\alpha$ may further enhance stability by preventing low-quality data points from having excessively large neighborhood radii.
>
> **Q3:  Effects and more insights of hyperparameter tuning.**
>
> > Does their choice ($\lambda$ and $\alpha$) significantly impact training stability or the final policy performance?
>
> (1) Effects of $\lambda$ and $\alpha$. We have conducted an ablation study in Section 4.4 on the Lagrange multiplier $\lambda$ and inverse temperature $\alpha$, analyzing learned Q-values and performance. ANQ generally exhibits stronger and more robust performance with intermediate values of $\lambda$ and $\alpha$. Since $\lambda=0$ and $\lambda=\infty$ correspond to no constraint and sample constraint respectively, the results indicate that the proposed neighborhood constraint outperforms both across a range of hyperparameters. Similarly, as $\alpha=0$ corresponds to a fixed neighborhood radius, the results show that advantage-based adaptive neighborhoods generally outperform uniform neighborhoods.
>
> > Could the authors provide more insight into how these hyperparameters are tuned in practice?
>
> (2) Hyperparameter details. The ANQ algorithm involves relatively little hyperparameter tuning. As detailed in Appendix C1, we fix $\alpha=1$ across all tasks, which yields consistently strong performance. For $\lambda$, we use only a small set of values: $\lambda = 5$ for Antmaze, and $\lambda \in \\{0.1, 5\\}$ for Gym locomotion, involving less tuning compared to baselines like CQL and SPOT.
>
> (3) Insights into hyperparameter selection. For selecting $\lambda$, a general principle is as follows: for narrow-distribution datasets (typically expert or demonstration data), a larger $\lambda$ can be used to induce smaller neighborhoods, thereby helping to more effectively control extrapolation error. For more dispersed datasets (often containing noisy or mixed-quality data), the impact of extrapolation error is relatively weaker; hence, a smaller $\lambda$ (i.e., larger neighborhoods) can be adopted to promote broader optimization over the action space and mitigate the impacts of suboptimal actions. Additionally, using a larger $\alpha$ in such cases can further enhance this effect by downweighting low-advantage actions more aggressively.
>
> > Is there an automatic way to tune these parameters based on the environment or data?
>
> (4) Automatic tuning. As with many offline RL algorithms, these parameters primarily control the trade-off between conservatism and exploration. Automatically tuning such parameters remains a well-recognized and important open problem in offline RL. Research progress in this direction could be integrated into our method by leveraging the above hyperparameter selection insights. While automatic tuning is not the main focus of this work, we believe it is a promising direction for future research.
>
> **Q4: While the method performs well in standard benchmark tasks, how does it scale to high-dimensional action spaces or environments with partially observable states?**
>
> Thanks for the good question.
>
> (1) Partially observable environments. To our knowledge, partially observable settings are not commonly featured in standard offline RL benchmarks. To simulate partial observability, we modify the Hopper and Walker2d environments by removing the agent's z-axis position, mimicking a height sensor failure. We compare representative algorithms with different constraint types, including TD3+BC, IQL, SPOT, and ANQ. Results averaged over 5 random seeds are shown in the table below. While all methods exhibit performance drops under partial observability, ANQ outperforms the baselines on most datasets and achieves the highest total score, demonstrating its robustness in such settings.
>
> Table 1: Normalized scores in partially observable environments without height information, averaged over 5 random seeds.
>
> | Dataset | TD3+BC | IQL | SPOT | ANQ |
> |---|---|---|---|---|
> | hopper-m-e | 85.3 | 65.7 | 82.3 | **99.6** |
> | hopper-m-r | 43.8 | 66.7 | 86.4 | **88.9** |
> | hopper-m | 55.9 | 54.1 | 50.6 | **67.0** |
> | hopper-r | 8.1 | 8.2 | 22.9 | **31.5** |
> | walker2d-m-e | 109.2 | 103.5 | 109.9 | **110.4** |
> | walker2d-m-r | 70.5 | 65.5 | **84.4** | 77.5 |
> | walker2d-m | 76.8 | 72.2 | **79.1** | 78.8 |
> | walker2d-r | 2.0 | 6.8 | 2.8 | **13.9** |
> | total | 451.5 | 442.5 | 518.3 | **567.6** |
>
> (2) Complex tasks and high-dimensional action spaces.
>
> AntMaze represents one of the most challenging offline RL benchmarks, with high-dimensional action space and sparse-reward, goal-reaching nature. The agent must not only coordinate multiple joints for locomotion but also compose suboptimal trajectory fragments to reach the goal. Empirical results show that ANQ significantly outperforms baseline methods on these tasks, especially in the most complex large-maze variants, demonstrating its effectiveness in such complex settings.
>
> From a method design perspective, the proposed neighborhood constraint may be particularly beneficial in high-dimensional action spaces. In such settings, support constraint methods often struggle to model the behavior policy accurately, while sample constraint methods risk being overly conservative due to limited coverage of near-optimal actions. ANQ avoids both issues by enabling controlled generalization without behavior policy modeling.
>
> **Reference**
>
> [1] Yu et al. Mopo: Model-based offline policy optimization. NeurIPS 2020.
>
> [2] An et al. Uncertainty-based offline reinforcement learning with diversified q-ensemble. NeurIPS 2021.
>
> [3] Chen et al. Decision transformer: Reinforcement learning via sequence modeling. NeurIPS 2021.
>
> [4] Wang et al. Diffusion policies as an expressive policy class for offline reinforcement learning. ICLR 2023.

---

> > ### Comment · Reviewer_JKyo · 2025-08-09
> >
> > Thank you for your detailed response. Your clarifications have convinced me, and I have decided to increase my score.

---

> > > ### Author Response · Authors · 2025-08-09
> > >
> > > Thank you very much for your positive evaluation and constructive feedback.

---

> ### Comment · Area_Chair_Vvee · 2025-08-06
>
> Please remember to respond to authors' rebuttal as soon as possible.
>
> Thank you!
>
> -AC

---

### Decision · Program_Chairs · 2025-09-17

**Decision:**

Accept (spotlight)

**Comment:**

The authors are addressing the important problem of constraining Q learning in offline reinforcement learning to stay within the data support and prevent extrapolation errors. They do so with a bilevel optimization scheme where the Q function is updated in the inner loop and the policy updated in the outer loop. They include a theoretical characterization of the extrapolation behavior of the Q function using NTK assumptions.

Strengths: The authors focus on an important problem of addressing extrapolation and regularization in the offline RL regime and have a novel and interesting approach. There are interesting theoretical results showing they are able to approximate the data support without requiring modeling the behavior policy directly.

Weaknesses: The method seems fairly complicated and analysis of how robust this method is compared to other, simpler offline RL methods is lacking. However, this has been addressed somewhat in the rebuttal with additional results based on hyperparameter sweeps shown.

Reasons for spotlight: I think this is an interesting direction and use of theoretical tools for addressing the extrapolation issue in offline RL. While there are weaknesses to the method and analysis, I believe the novelty of the approach outweigh these.

Reviewers mainly had concerns about the stability of the bilevel optimization process, the sensitivity to hyperparameters, and clarity of the main contribution and readability of the paper. Most of these concerns were addressed in the rebuttal phase, and I encourage the authors to continue improving on clarity of the contributions for the camera ready.